# Rapid single-cell physical phenotyping of mechanically dissociated tissue biopsies

Despina Soteriou[1,8], Markéta Kubánková[1,8], Christine Schweitzer[1],
Rocío López-Posadas[2,3], Rashmita Pradhan[2,3], Oana-Maria Thoma[2,3,4],
Andrea-Hermina Györfi[3,5], Alexandru-Emil Matei[3,5], Maximilian Waldner[2,3,4],
Jörg H. W. Distler[3,5], Stefan Scheuermann [6], Jens Langejürgen [6],
Markus Eckstein[4,7], Regine Schneider-Stock [4,7], Raja Atreya[2,3,4],
Markus F. Neurath [2,3,4], Arndt Hartmann[4,7] & Jochen Guck [1] ✉

During surgery, rapid and accurate histopathological diagnosis is essential for clinical decision making. Yet the prevalent method of intra-operative consultation pathology is intensive in time, labour and costs, and requires the expertise of trained pathologists. Here we show that biopsy samples can be analysed within 30 min by sequentially assessing the physical phenotypes of singularized suspended cells dissociated from the tissues. The diagnostic method combines the enzyme-free mechanical dissociation of tissues, real-time deformability cytometry at rates of 100–1,000 cells s⁻¹ and data analysis by unsupervised dimensionality reduction and logistic regression. Physical phenotype parameters extracted from brightfield images of single cells distinguished cell subpopulations in various tissues, enhancing or even substituting measurements of molecular markers. We used the method to quantify the degree of colon inflammation and to accurately discriminate healthy and tumorous tissue in biopsy samples of mouse and human colons. This fast and label-free approach may aid the intra-operative detection of pathological changes in solid biopsies.

Changes in physical properties of cells, such as cell size, shape or deformability, are pivotal to the pathology of some diseases and hold great potential as a diagnostic or prognostic marker[1,2]. In the past decades, a variety of tools have been developed to examine the mechanical properties of cells, including micropipette aspiration, atomic force microscopy, microbead rheometry and optical traps[3,4]. The field has seen an exponential increase in publications that suggest a strong correlation between cell mechanical phenotype and disease state, including sepsis[5,6], malaria[7], diabetes[8], sickle cell anaemia[9] and cancer[10–12].

Unfortunately, these conventional techniques suffer from low cell throughput and the requirement of deep specialist knowledge for operation, which limits their use as a diagnostic tool. Real-time fluorescence and deformability cytometry (RT–FDC)[13,14] is one of several new microfluidic techniques[10,15–22] that have overcome these drawbacks, allowing the assessment of physical properties of single cells in a label-free and high-throughput manner, opening a new avenue to clinical diagnostics. RT–FDC is not only fast (with up to 1,000 cells analysed per second), but in addition to cell deformability it also provides

[1]Max Planck Institute for the Science of Light and Max-Planck-Zentrum für Physik und Medizin, Erlangen, Germany. [2]Department of Medicine 1— Gastroenterology, Pneumology and Endocrinology, Friedrich-Alexander-University Erlangen-Nürnberg (FAU) and University Hospital Erlangen, Erlangen, Germany. [3]Deutsches Zentrum für Immuntherapie (DZI), Friedrich-Alexander-University Erlangen-Nürnberg (FAU) and University Hospital Erlangen, Erlangen, Germany. [4]Comprehensive Cancer Center Erlangen-EMN (CCC ER-EMN), Erlangen, Germany. [5]Department of Internal Medicine 3— Rheumatology and Immunology, Friedrich-Alexander-University Erlangen-Nürnberg (FAU) and University Hospital Erlangen, Erlangen, Germany. [6]Clinical Health Technologies, Fraunhofer IPA, Mannheim, Germany. [7]Institute of Pathology, University Hospital, Friedrich-Alexander University Erlangen-Nürnberg (FAU), Erlangen, Germany. [8]These authors contributed equally: Despina Soteriou, Markéta Kubánková. ✉e-mail: jochen.guck@mpl.mpg.de

multi-dimensional information obtained directly from cell images. The diagnostic potential of RT–FDC has been demonstrated in many human disease conditions ranging from leukaemia to bacterial and viral infections including coronavirus disease 2019 (refs. 23–27). However, until now, the applicability of the technique was limited to analysing cultured cells or liquid biopsies from blood or bone marrow.

Solid tissue biopsy is the most common method for characterizing malignancy and is fundamental in guiding surgeons during intra-operative and peri-operative management of cancer patients. Diagnostic assessment of solid tissue biopsies is commonly delivered through intra-operative consultation pathology, which relies on histopathological analysis of frozen biopsy sections[28]. The conventional workflow of intra-operative diagnosis involves numerous processing steps, staining reagents and the microscopic inspection of tissue slices by experienced pathologists for expert analysis. Moreover, sample preparation is time-, resource- and labour-intensive. Alternative workflows have been proposed[28], including stimulated Raman spectroscopy[29,30], optical coherence tomography[31] and fluorescence microscopy[32,33], but have not yet been implemented. The need for an approach that reduces sample preparation and time to diagnosis is therefore imminent.

In this Article, we present a rapid, label-free diagnostic method for solid tissue biopsies. The approach combines the enzyme-free, mechanical dissociation of tissues using a tissue grinder (TG) for the quick and simple isolation of viable single cells[34,35] with the sequential assessment of cellular physical phenotypes of thousands of individual cells using RT–FDC. First, we screen a panel of different mouse tissues and assess the cell yield, viability and the feasibility of RT–FDC measurement upon the mechanical dissociation of tissue. We illustrate the ability to distinguish subpopulations of tissue cells purely based on the image-derived physical parameters without prior knowledge or additional molecular labelling, which can enhance conventional flow cytometry, which relies on multi-colour panels of markers for identifying cells. We also show that our approach can determine inflammatory changes in colon tissue, based on the measurement of cell deformability in the microfluidic system. Moreover, we examine frozen and fresh biopsy samples from mouse and human colon and show that RT–FDC can distinguish healthy from cancerous tissues, by using principal component analysis (PCA) and machine learning on the multi-dimensional data. The findings demonstrate that assessing the physical phenotype of tissue-derived single cells using RT–FDC is an alternative strategy to detect an inflammatory or malignant state. Our procedure, which can deliver results within 30 min, has potential as an intra-operative diagnostic pipeline to sensitively detect pathological changes in biopsies and, more generally, to identify and characterize cell populations in tissues in an unbiased and marker-free manner.

## Results

### Physical phenotyping of cells from mechanically dissociated tissues

Before assessing the physical phenotype of cells, the first challenge faced was the quick extraction of single cells from solid tissues on a timescale of minutes, while aiming for a maximally accurate representation of the heterogeneity of cell subpopulations. For this, we used a TG, a mechanical dissociation device based on counter-rotating rows of grinding teeth (Fig. 1) assembled into a Falcon tube[35]. The device automatically executes a predefined sequence of alternating cutting and grinding steps to isolate single cells from a solid tissue. In total, ten different murine tissues were processed using either TG or conventional enzymatic protocols for comparison (Supplementary Tables 1 and 2). Viability was 70–90% in most tissues; cell yield was similar to enzymatic dissociation and tissue dependent (Extended Data Fig. 1). The key advantage of mechanical dissociation was that the processing time took less than 5 min per sample, as opposed to tens of minutes or even several hours for the enzymatic protocols. The speed of the

extraction presumably helps to preserve biochemical and biophysical phenotypes in conditions close to those in situ.

Next, the extracted single cells were analysed using RT–FDC. In an RT–FDC measurement, hundreds of cells per second, suspended in a high-viscosity methyl cellulose buffer, are pushed through a microfluidic channel constriction, where they are deformed by shear stress and pressure gradients and an image of each cell is obtained. Several physical parameters were calculated from the images in real time, namely deformation, cell size, brightness, standard deviation of brightness, aspect ratio and area ratio (for details, see Supplementary Table 3). Additionally, the fluorescence module[14] was used to detect the expression of cell surface markers.

Illustrative examples of the distribution of physical parameters of cells extracted from liver, colon and kidney are shown in Fig. 2. Each of these clusters was composed of cells with similar physical phenotype (gated according to the density plot) and surface marker expression (Extended Data Fig. 2). For example, a cluster of cells with similar physical properties (in this case defined by average brightness and cell size) was mainly composed of epithelial cell adhesion molecule (EpCAM)-positive cells (Fig. 2a), demonstrating that a clean population of epithelial cells can be distinguished in a label-free manner, purely using image-derived physical parameters.

Figure 2b,c illustrates the advantage of using image-based physical phenotyping in addition to using conventional fluorescence-based flow cytometry alone. In conventional flow cytometry it is hardly possible to distinguish individual subpopulations of epithelial (EpCAM+) cells unless extra panels of fluorescent antibodies against known and pre-defined cell types are used. Distinction of various subpopulations was possible with RT–FDC owing to the additional information depth provided by physical phenotype parameters. Within the epithelial cells of colon, we identified seven clusters of cells purely based on brightness and size (Fig. 2b). Similarly, within the leukocyte (CD45+) population of the kidney, we found four different clusters based on the cell size and deformation parameters (Fig. 2c). We note that, using the sorting modality recently developed for RT–FDC[36], any of these cell populations can be isolated according to the image-derived parameters and analysed for their molecular identity, for example, by subsequent RNA sequencing.

RT–FDC can also be used to capture cell interactions. Using the aspect ratio and cell size parameters, we identified cell doublets in thymus, spleen and kidney samples. Many doublets were composed of two different cell types, according to the cell surface markers (Extended Data Fig. 3). The position of the cell within the channel in combination with the position of the fluorescence peak allowed us to identify, for instance, that a doublet was composed of a leukocyte (CD45+) and an endothelial cell (CD31+) (Extended Data Fig. 3b,d,f). Using the RT–FDC sorting module[36], cell doublets can be isolated label free for further molecular analysis and downstream applications, including studies of physically interacting immune cells in tissue[37].

An important question to consider when using mechanical dissociation of tissues and label-free analysis by physical phenotype is whether this approach faithfully represents the distribution of cell types present in the tissue. While this is impossible to assess for all tissues and applications in general, it is instructive to have a closer look at liver as a specific tissue (Extended Data Fig. 4a–c). Mechanical dissociation seems less disruptive to sensitive cells such as hepatocytes, which are prone to cell death and often lost during standard isolation procedures[38]. Upon dissociation of murine liver tissue, cells above 150 μm² in cross-sectional cell area (~7 μm radius) were determined as hepatocytes according to their morphology and size[39]. As the major parenchymal cell type of the liver, hepatocytes account for 70% of the liver cell population and take up nearly 80% of liver volume[40]. In the cell suspension obtained using TG, the proportion of hepatocytes to total cells was on average 52.5%, much closer to the real representation in tissue compared with the 7.7% for

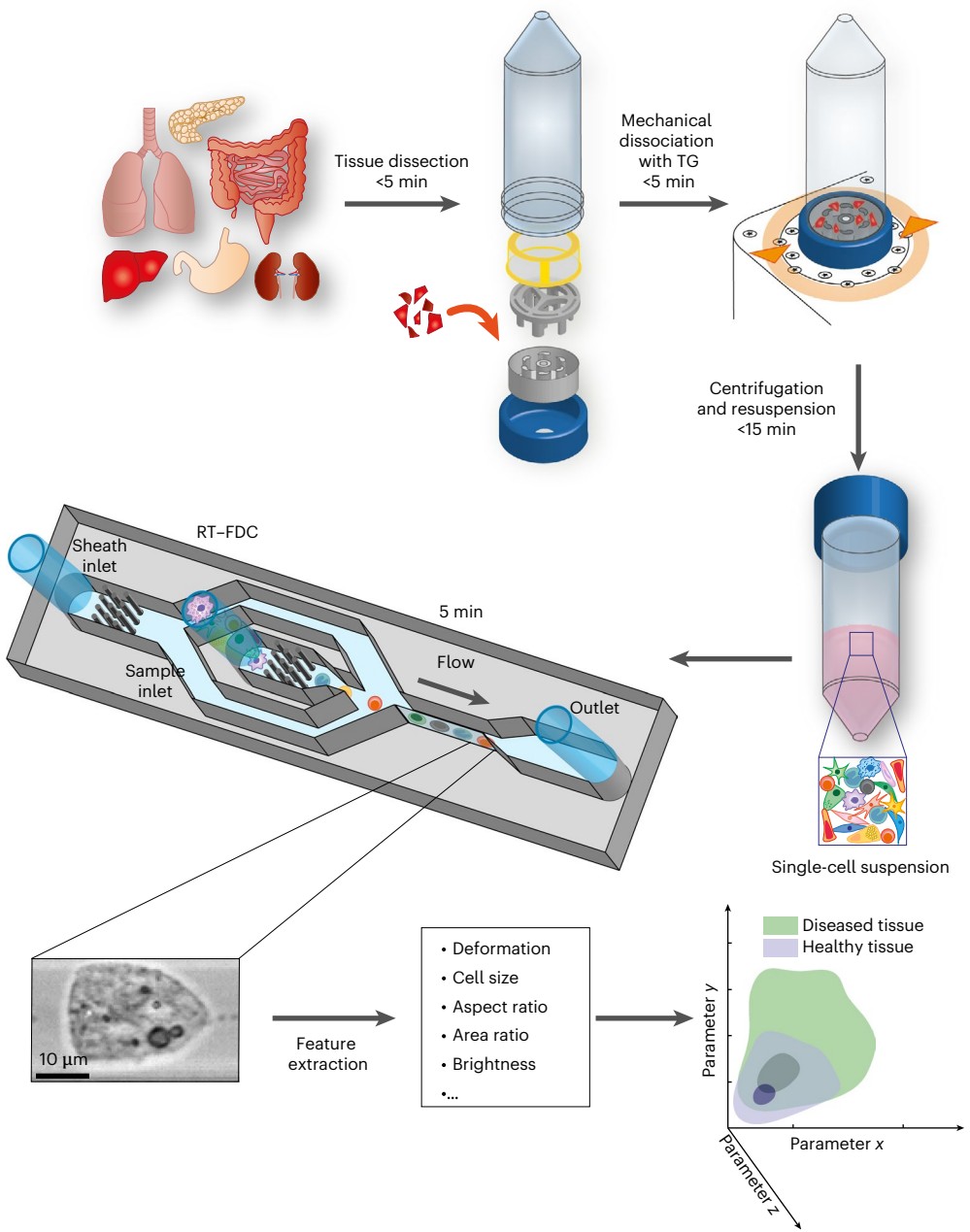

**Fig. 1 | Schematic of the physical phenotyping method.** The tissue sample is dissected in small pieces and placed into the inner rotor of the TG unit containing culture medium[35]. Mechanical dissociation is performed by a pre-programmed, automatically executed sequence of clockwise and counter-clockwise rotations. Dissociated cells are centrifuged and resuspended in measuring buffer. The sample is loaded onto a microfluidic chip and analysed using RT−FDC. A brightfield image of every single one of typically 10,000 cells is captured. Various features are extracted from the images, which are used for multi-dimensional analysis. In total, the procedure from tissue to result takes less than 30 min.

enzymatic digestion. Moreover, distinct subpopulations of hepatocytes could be identified according to cell size. We hypothesize that these populations correspond to hepatocytes of differing ploidy, as DNA content is strongly correlated with cell volume[41]. If confirmed, for example, by correlation with a quantitative fluorescence analysis of DNA amount in each cell, our method could serve as a tool for the label-free monitoring of ageing and pathophysiological processes in the liver, which are linked with the proportion of polyploid hepatocytes[42]. In other tissues, such as lung, the differences between mechanical and enzymatic dissociation were not as prominent and neither technique gave a bias towards a specific cell population (Extended Data Fig. 4d). However, for the general applicability of our approach to the diverse possibilities it opens up for tissue analysis, of course, further experimental work is needed.

There are specific−in particular diagnostic−applications of our approach where the faithful determination of cell numbers originally present in the tissue in situ is less important. After all, the physical phenotypes detected also reflect diverse cellular responses to the sample processing, the adhesion of cells to each other and to the extracellular matrix, and the connectivity and mechanical strength within the tissue. All of these aspects can be altered in pathological conditions and would be picked up by our approach. We demonstrate the diagnostic utility in two specific clinically relevant use cases related to the colon.

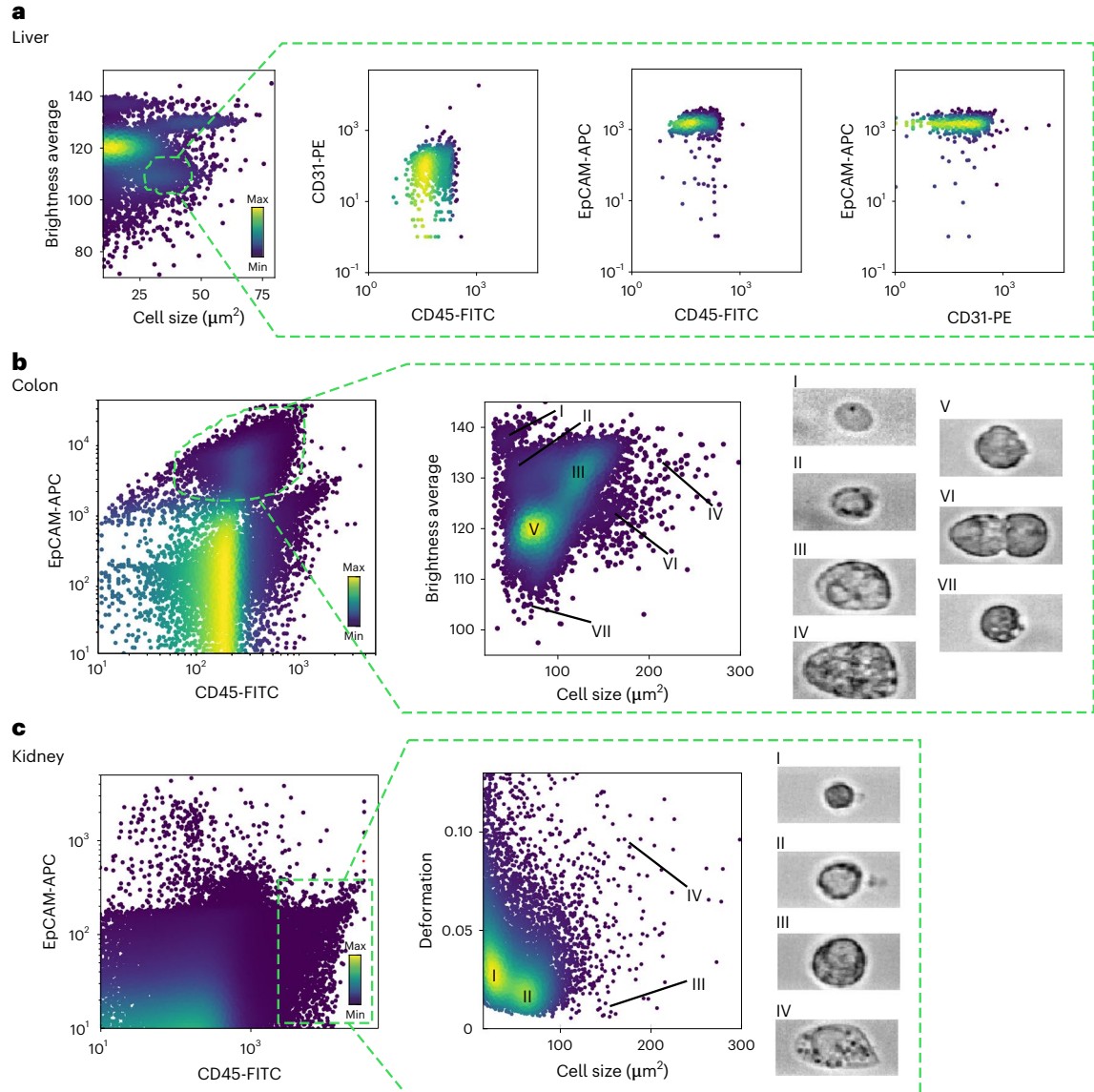

**Fig. 2 | Illustrative scatter plots of physical parameters of cells from murine liver, colon and kidney samples. a**, Illustrative scatter plot of brightness average versus cell size for cells isolated from the liver showing numerous clusters of cells. The marked population of cells (forming a cluster of cell size 25–50 µm² and brightness average 100–115) is enriched for EpCAM-positive (epithelial) cells but devoid of CD31-positive (endothelial) cells or CD45-positive cells (leukocytes). FITC, fluorescein isothiocyanate; PE, phycoerythrin; APC, allophycocyanin.

**b**, Illustrative scatter plots of colon cells stained for EpCAM and CD45 cell surface markers. Within the EpCAM-positive population, seven subpopulations of cells can be identified on the basis of density plots of physical parameters, such as brightness and cell size. **c**, Illustrative scatter plots of kidney cells stained for EpCAM and CD45. Within the CD45 population, four subpopulations are identified based on similarities in cell size and deformation. The colour map in the scatter plots represents the event density.

## Tissue inflammation is detected by cell physical phenotyping

Inflammatory bowel diseases (IBD), such as Crohn's disease and ulcerative colitis, are chronic inflammatory disorders of the intestine associated with a compromised epithelial/mucosal barrier and activation/recruitment of immune cells[43]. Although the aetiology of IBD is still not fully understood, much of our understanding about IBD comes from experimental animal models of intestinal inflammation. One such model is adoptive transfer of naïve T cells into Rag1-deficient mice to induce experimental colitis (T-cell transfer model of chronic colitis, from here on referred to as transfer colitis). The severity is then commonly quantified via a histopathological score generated from haematoxylin and eosin (H&E)-stained slides of the colon tissue.

Our goal was to investigate changes in the physical phenotype of colon cells during transfer colitis. Scatter plots of deformation versus cell size suggest a difference between disease and healthy tissue (Fig. 3a), where cells from disease tissue appear less deformed than cells from healthy tissue. Upon examination of the CD45+ cells (Fig. 3b,c), it became evident that the transfer colitis samples were characterized by a high abundance of leukocytes with low deformation, probably lymphocytes. Overall, we found a significant decrease of median deformation with strong effect size in the transfer colitis samples, accompanied by a significant increase in the percentage of leukocytes, in accordance with infiltration of adoptively transferred lymphocytes (N = 14; Fig. 3d). The median deformation of cells was strongly negatively correlated with the percentage of leukocytes, with a Pearson's correlation coefficient of r(12) = −0.69 (P = 0.0065; Fig. 3e). Furthermore, the median values of cell size and deformation were linked with expert H&E scoring (Supplementary Fig. 1); although

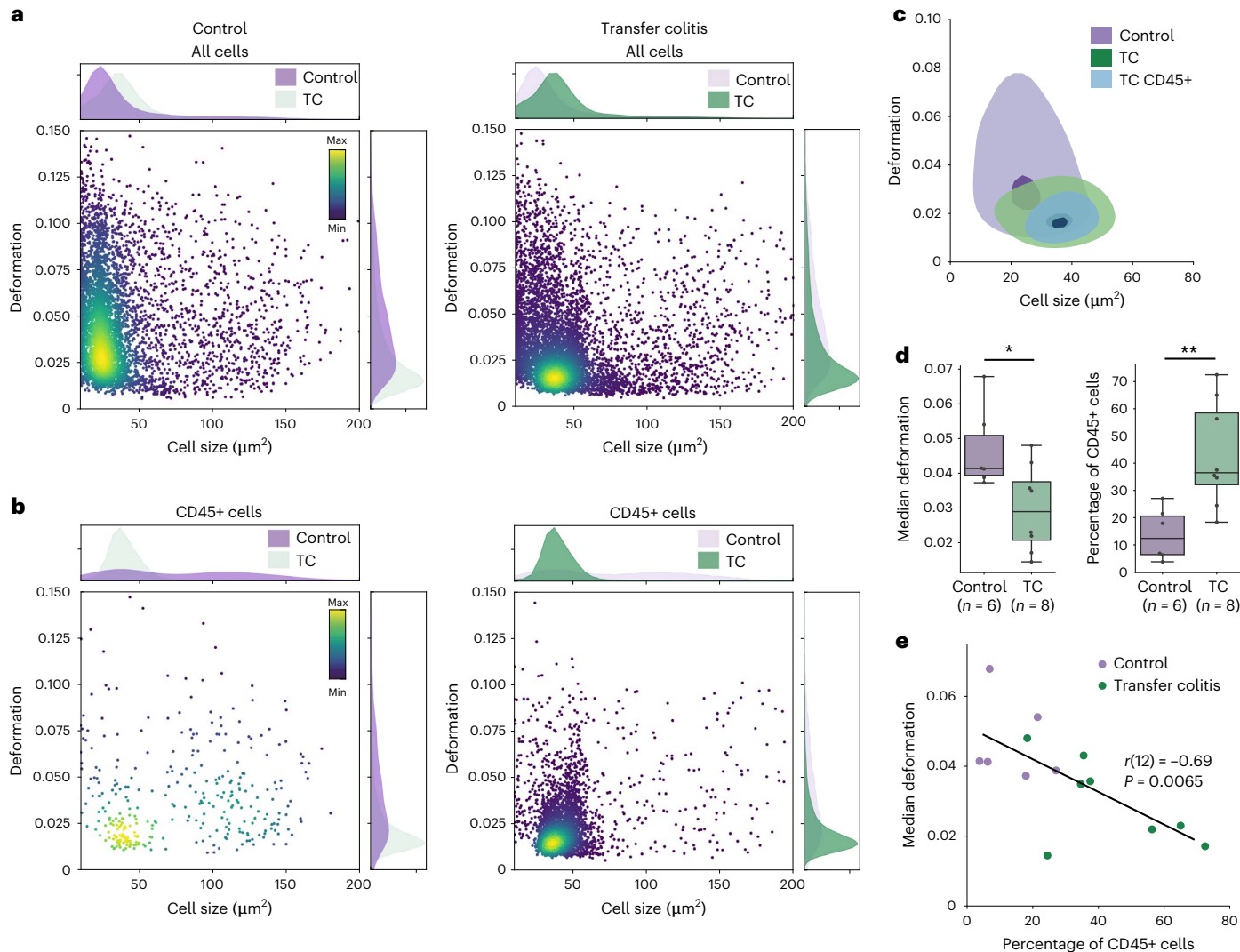

**Fig. 3 | Physical phenotyping of cells via RT–FDC reflects tissue inflammation.** **a**, Cell size versus deformation scatter plots of cells isolated from transfer colitis tissue samples (TC) compared with healthy murine colon tissue (Control); with corresponding cell size and deformation histograms. **b**, The same two colon samples gated for CD45-positive cells, showing the enrichment of leukocytes in transfer colitis samples, accompanied by changes of the physical phenotype parameters. The colour map in the scatter plots represents the event density. **c**, Kernel density estimate plots of samples shown in **a** and **b**, with contours marking the 0.5 (light shade, outer contour) and 0.95 (dark shade, inner contour)

levels. **d**, Quantification of median deformation and percentage of CD45-positive cells ($n = 14$ biologically independent animals over three independent experiments). Boxes extend from the 25th to the 75th percentile with a line at the median; whiskers span 1.5× the interquartile range. Statistical comparisons were performed using two-sided Mann–Whitney $U$ test; median deformation *$P = 0.0227$ and $r = 0.55$, % CD45+ **$P = 0.0041$ and $r = 0.7$ ($r$, effect size). **e**, Two-sided Pearson's correlation of median deformation of all cells and the proportion of leukocytes (CD45+ cells); $P = 0.0065$ and $r = -0.69$.

correlation via linear fitting was not possible. Transfer colitis samples with high H&E score exhibited bigger cell size and lower deformation compared with healthy tissue. A noteworthy observation was that the healthy tissue was more difficult to mechanically break apart into single cells than the diseased tissue, which yielded more events for analysis.

Our observations that the physical phenotype of cells changes upon inflammation, together with the growing evidence that chronic inflammation is associated with malignancy[44,45], led us to speculate that our approach might detect changes in biopsy samples from tumours. We confirm this possibility for both mouse and human samples.

**Distinction of tumour and healthy tissue in mouse colon**

Previous studies have found differences between the mechanical properties of cancer cells and their healthy counterparts[11,12,46–49]. A major drawback of these studies is laborious sample preparation and low

measurement throughput that limits the conversion of these studies to actual diagnostic approaches. Given the rapidity of our approach to obtain and assay the mechanical phenotype of single cells from solid tissues, we explored its potential to detect colorectal cancer. We used mice deficient in an intestinal epithelial cell-specific protein with a key role in epithelial integrity. These animals spontaneously develop colon tumours. We examined a total of 16 mice and compared cells isolated from tumours with cells from a healthy part from the colon of the same animal. We analysed cells greater than 60 μm$^2$ (determined by cross-sectional area), as below this threshold the sample was comprised mainly of immune cells and small debris (Supplementary Fig. 2).

Our results showed that the physical phenotype of cells from tumour tissue significantly differed from the control samples. Representative plots from a single mouse in Fig. 4a–c demonstrate that cells from the tumour had larger cell size and higher deformation than

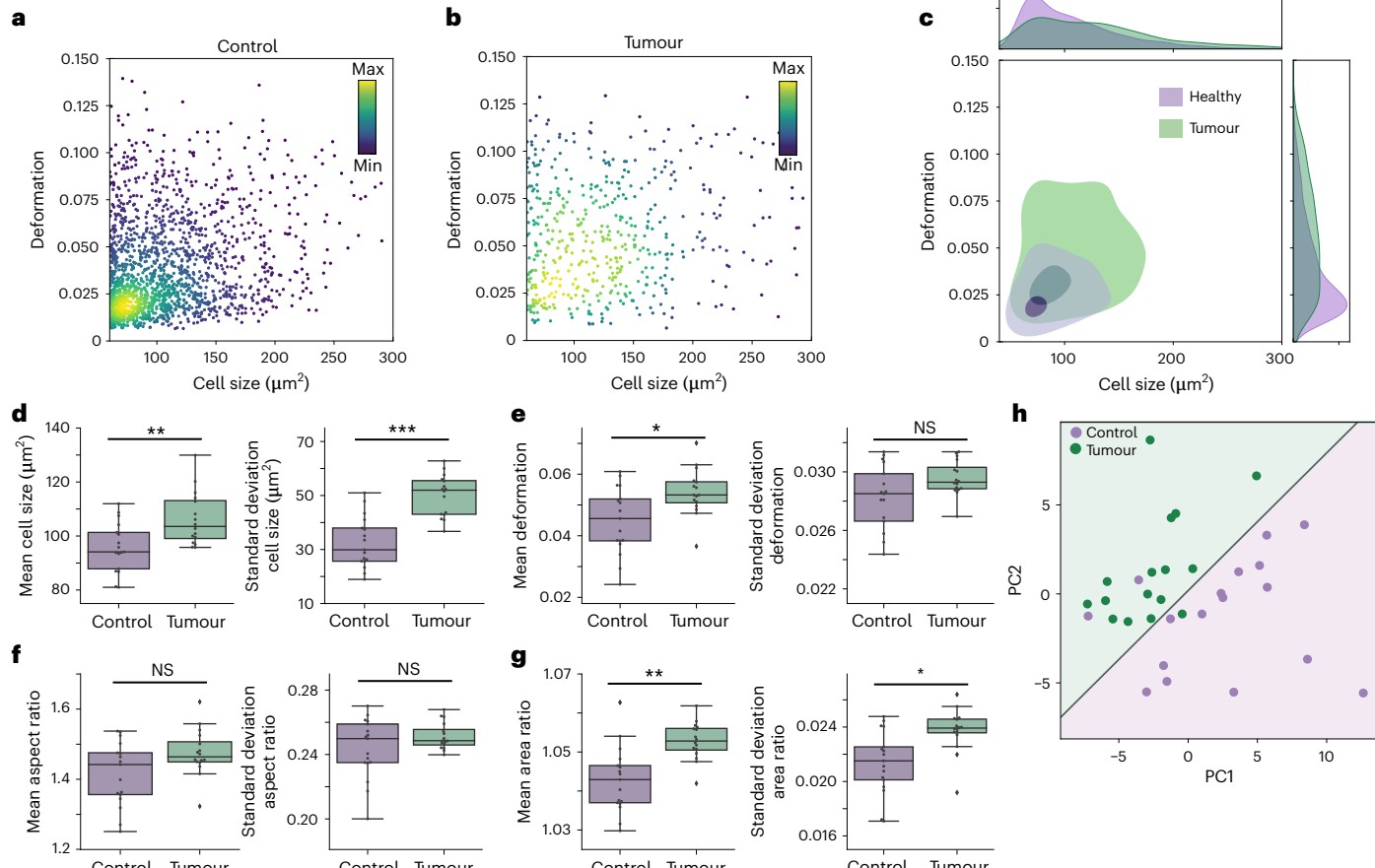

**Fig. 4 | Physical phenotyping of cells in tumour and healthy mouse colon tissue. a,b,** Cell size versus deformation scatter plots of a control sample (**a**) of murine colon tissue compared with tumour tissue (**b**). The colour map in the scatter plots represents the event density. **c,** Kernel density estimate plots of samples shown in **a** and **b**, with contours marking the 0.5 (light shade, outer contour) and 0.95 (dark shade, inner contour) levels. The cell size and deformation histograms demonstrate greater heterogeneity of cell size and deformation in tumour (green) compared with the control tissue (purple). **d–g,** Means and standard deviations of physical phenotype parameters of 16 control (purple) and 16 tumour samples (green) (n = 16 biologically independent animals over six independent experiments). Boxes extend from the 25th to the

75th percentile with a line at the median; whiskers span 1.5× the interquartile range. Statistical comparisons were performed using two-sided Wilcoxon signed rank test; r, effect size: cell size (**P = 0.0019, r = 0.55), standard deviation of cell size (***P = 0.0005, r = 0.61) (**d**); deformation (*P = 0.026, r = 0.39), standard deviation of deformation (NS, not significant) (**e**); aspect ratio (NS), standard deviation of aspect ratio (NS) (**f**); area ratio (**P = 0.0023, r = 0.54), standard deviation of area ratio (*P = 0.013, r = 0.44) (**g**). **h,** PCA of mouse colon tissue samples, where green points represent tumour samples and purple points represent the control samples. Linear regression analysis was performed on PC1 and PC2 with the resulting two categories shown as purple (control) and green (tumour) background colours.

their healthy counterparts. The analysis of all 32 samples revealed that cells from tumours had significantly higher mean cell size (Fig. 4d), deformation (Fig. 4e) and area ratio (Fig. 4g), with moderate to strong effect sizes. The tumour samples also exhibited greater heterogeneity, demonstrated by the broader distribution in Fig. 4c and significantly higher standard deviations of cell size and area ratio (Fig. 4d,g).

We next investigated whether the physical phenotype differences could be exploited for the reliable distinction between tumorous and healthy tissue. For this, we divided cells into three categories according to cell size (60–90, 80–120 and 120–400 μm²). For each size category, 12 parameters were derived: mean, median and standard deviations of the cell size, deformation, aspect ratio and area ratio; adding up to a total of 36 parameters for each sample (Supplementary Fig. 3). These parameters were used for PCA, Fig. 4h. The two principal components, PC1 (39.8%) and PC2 (18.7%), explained 58.5% of the variance. The relative importance of physical features in determining the principal components is shown in Supplementary Fig. 3. The most dominant feature for PC1 was the deformation of cells between 60 and 120 μm², whilst in the case of PC2, cell size parameters prevailed. Logistic regression performed on the PCA (shown by the linear divide in Fig. 4h) demonstrated

that the condensed physical phenotype information represented by the principal components suffices to distinguish between healthy and tumour tissue; 29 out of 32 samples lay in the correct region. Finally, we analysed the correlation between deformation and cell size and found it to be weak or non-existent (Supplementary Fig. 4). This led us to conclude that deformation and cell size were independent predictors of tumours in murine colon samples, further demonstrating the added value of deformation measured via RT–FDC as a diagnostic marker.

**Distinction of tumour and healthy tissue in human biopsies**
We next sought to challenge our method for detecting tumours from human biopsy samples. As a first step, we performed RT–FDC analysis on cells isolated from 13 cryopreserved biopsy samples of colorectal cancer and 13 samples of healthy surrounding tissue from the same patients. PCA was performed on 45 parameters (Fig. 5a and Supplementary Fig. 5) with 41.7% of the variance explained by the two principal components (25.3% and 16.4% for PC1 and PC2, respectively); the selection of RT–FDC parameters was optimized to obtain a good separation between the healthy and tumour tissue. The PCA showed that tumour and healthy samples segregated well along PC2, mainly by the

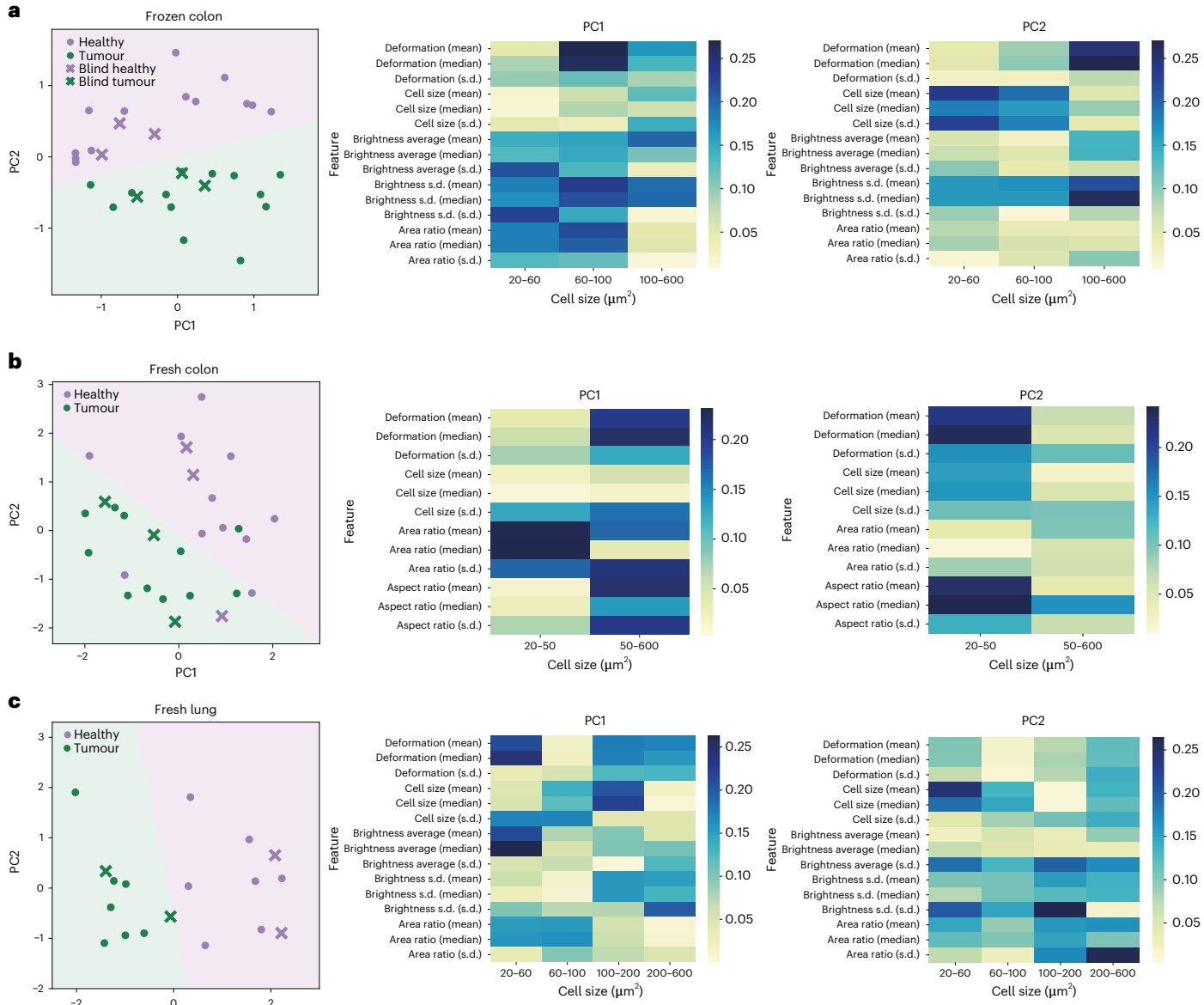

**Fig. 5 | Distinction of tumour and healthy tissues in human biopsies using PCA and logistic regression.** In the PCA plots on the left, each green point represents a tumour sample from one patient; purple points represent the corresponding healthy surrounding tissue from the same patient. Logistic regression was performed on each of the PCAs with the resulting two categories shown as purple (healthy) and green (tumour) background colours. Crosses represent blind experiments used for the validation of the trained model. The feature importance analysis to the right of the PCA plot shows the colour-coded significance of each feature for determining PC1 and PC2 for that particular tissue; the x axis lists cell size categories; the y axis lists RT–FDC parameters and their statistical features derived across cells in the corresponding size category (in brackets). s.d., standard deviation. **a**, PCA of RT–FDC parameters of 32 frozen colon samples (16 tumour biopsies and 16 samples of healthy surrounding tissue; n = 16 biologically independent samples over 16 independent experiments). **b**, PCA of RT–FDC parameters of 28 fresh colon biopsy samples (14 tumour, 14 healthy; n = 14 biologically independent samples over 14 independent experiments). **c**, PCA of RT–FDC parameters of 18 fresh lung biopsy samples (9 tumour, 9 healthy; n = 9 biologically independent samples over 9 independent experiments).

deformation and standard deviation of brightness of cells larger than 100 µm² (Fig. 5a). Cell size parameters of cells below 100 µm² also contributed to the separation of the samples. Excluding the most important parameter (deformation of cells larger than 100 µm²) resulted in worse separation between healthy and tumour samples (Supplementary Fig. 6). Logistic regression was performed on the PCA (shown by the linear divide in the PCA plot) and used to predict the classification of six blind samples (shown as crosses in Fig. 5a); all six samples were correctly classified as either healthy or tumour tissue, respectively. We examined the minimal number of cells needed for correct classification of these blind samples (Supplementary Fig. 7). Approximately 1,500 cells from

a sample had to be analysed for correct classification, corresponding to RT–FDC measurement time of approximately 5 min.

The short combined processing and analysis time (<30 min) and the positive results obtained on frozen tissue biopsy sections suggest the use of the method for intra-operative pathology–the examination of a patient's biopsy sample during surgery. To explore whether even the freezing step could be omitted, we analysed freshly excised biopsies from colorectal cancer patients (N = 14). The algorithm trained on frozen colon tissue did not perform well for fresh tissue, possibly due to differences of physical phenotype between the frozen and fresh tissue (Extended Data Fig. 5). Therefore, a new PCA was performed

on data from fresh colon biopsies in two size categories (20–50 and 50–600 μm$^2$) (Fig. 5b), where 68.5% of the variance was explained by the two principal components (36.5% and 32% for PC1 and PC2, respectively). Here, the deformation of cells contributed strongly in the PCA and cell size was less important than the other physical phenotype parameters. Upon logistic regression, only 3 out of 22 samples used for PCA and 1 out 6 of the blind samples were not correctly classified, which could be attributed to inter-tumour or intra-tumour heterogeneity. Nevertheless, using our approach on blind samples we achieved 100% accuracy in classifying healthy and tumour samples from frozen biopsies, and 83% accuracy for fresh biopsy samples.

To validate our method on tissue from a different organ, we applied it to freshly excised lung biopsy samples from nine cancer patients. PCA combined with logistic regression readily separated seven healthy from seven tumour samples and further four blind samples were correctly classified (Fig. 5c). In the PCA, 46.9% of the variance was explained by PC1 and PC2 (31.2% and 15.7%, respectively). Deformation parameters contributed strongly to PC1, which again demonstrates that the unique information brought by the cell deformability measurement is useful for distinguishing between tumour and healthy tissue.

We also tested the sensitivity of our approach to the situation where only few cancer cells are present (tumours with low tumour cellularity and extensive desmoplastic tumour stroma content) or remain (tumours following chemo- or radiochemotherapy with nearly complete remission) in the tissue samples available. This aspect is especially important for clinical situations where intra-operative analysis is used to determine whether the operative margin is free of cancer (so called, frozen sections), which can be particularly difficult when only very few tumour cells are present. We performed an experiment in which we analysed a mixture of fresh tumour and healthy lung tissue samples at different ratios (Supplementary Fig. 8). A sample consisting of 50% healthy and 50% tumour tissue was classified as a tumour sample. Of course, not all the tumour tissue consists of cancer cells, so that the real sensitivity to detect cancer cells is higher than apparent here. In fact, some of the colon tumour samples had relatively high stromal content. In the extreme cases (Supplementary Table 5, frozen colon tissue samples 7 and 11) the stromal content was 98% and 80%, meaning the patients had nearly no residual tumour after neoadjuvant radiochemotherapy. Still, these samples were correctly classified as tumour. This result is remarkable as it points out a possible solution to the sampling problem present in conventional histopathological analysis—especially in frozen section scenarios. The result of the latter very much depends on whether the pathologist inspects and selects the correct tissue location where cancer cells are still present. Due to time constraints and technical limitations of slide preparation in frozen section scenarios, only a small fraction of the total resected tissue specimen can be visualized. If the dissociation of the tissue into single cells, and the analysis of a random subset of these, can sensitively detect the presence of as low as 20%—or even 2%—of cancer cells present in a given tissue sample, this would be a clear advance over the state of the art. More specific research is needed to firmly establish this. Finally, our method can also detect low-grade cancer, where any differences in physical parameters are expected to be more difficult to detect than in high-grade cancer. The vast majority of the analysed samples were G2 (moderately differentiated) or G3 (poorly differentiated/undifferentiated, often also referred to as 'high grade'; Supplementary Tables 5–10). In the lung tissue dataset, one of the samples was classified as the lowest-grade G1 (well differentiated; Supplementary Table 10). Therefore, the method is not limited to high-grade cancer.

## Discussion

We have shown a quick and simple method for the processing and analysis of cells from solid tissue, suitable for biopsy-based diagnostics. Mechanical dissociation of tissue is followed by high-throughput analysis of cells in deformational flow. Within a few minutes, thousands of cells are imaged, and various physical phenotype features are extracted from each cell image. The method is label free and relies simply on brightfield images, in contrast to molecular diagnostic tools or conventional flow cytometry, where expensive reagents or fluorescent markers are needed. Importantly, the information is available within 30 min of biopsy excision, which can be an advantage when there is necessity to detect pathology quickly. This is the case during intra-operative consultation providing diagnostic information during cancer surgery and often defining the further course of the procedure. The standard workflow requires transport of the biopsy sample to the pathology department, where it is embedded in a mounting medium (optimal cutting temperature compound), frozen and cut in thin slices using a cryostat. The slides are then prepared with H&E staining, and pathologists assess numerous characteristics including the nature of the lesion (that is, its malignancy) using a microscope[30,50]. Our workflow circumvents the freezing and staining steps, could be performed directly on-site, and allows to detect malignancy on the basis of the automated assessment of physical parameters of single cells.

Beyond intra-operative diagnosis, we show that the method is useful for the rapid examination of IBD samples. Clinical diagnosis of IBD in most cases requires the combination of different tests, including a blood test, stool examination, endoscopy and histological analysis of mucosal biopsies[51,52]. Histological scoring has growing importance in IBD, as the histological level of inflammation correlates with recurrence of disease, probability of surgery and risk of cancer. We show that the degree of tissue inflammation in a colon biopsy sample can be obtained by monitoring the physical phenotype of the cells via RT–FDC, bypassing the need for staining or expert assessment. We envision that the method could be used to monitor temporal inflammatory changes to assess disease progression and response to treatment, and to provide an objective diagnostic scoring system for daily clinical practice, which is currently lacking for IBD.

Previous studies on cancer cells have shown a strong correlation between malignancy and the mechanical properties of cells[10–12,46,49,53]. Here we exploit this correlation for detecting malignancy in human tissue biopsies. RT–FDC probes cell deformability, at a high-throughput rate, by exposing cells to shear flow in a microfluidic channel; and it allows for the mechanical phenotyping of single cells, using an analytical model and numerical simulations[54,55]. Assuming an initial spherical cell under normal (stress-free) conditions, RT–FDC can provide an elastic modulus as a quantitative measure of cell stiffness. However, in heterogeneous tissue samples, such as the ones used in this study, the cells are often not spherical before entering the microfluidic channel and an elastic modulus cannot be obtained. Nevertheless, the degree of deformation in this standard deformation assay can be interpreted as a qualitative measure of deformability and the deformation information inherent in the images is shown to be valuable diagnostic marker. PCA of murine colon samples and human colorectal biopsies revealed that cell deformation in standardized channel flow conditions is key for distinguishing between healthy and tumorous tissue in the examined biopsy types. This highlights the uniqueness of the information brought by this method, currently missing from routine diagnostic practices that, so far, rely mostly on histological assessment. Following this proof-of-concept study, it will be necessary to investigate whether the method can be adapted to different types of cancer or tissue. We expect that cell deformability changes might manifest more in certain types of cancer than in others. There may be certain application areas where the method has potential for improving diagnostic practice.

For practical clinical use, it will be beneficial to integrate the tissue-processing and single-cell-phenotype analysis into a single automated pipeline. Although mechanical dissociation using a TG is an efficient way to obtain single cells from tissues for diagnostic applications, it will be important to reduce the manual handling steps, such as filtering and concentrating cells. However, even in its current state, it is faster and more cost effective than enzymatic processing

of tissue. A key advantage of mechanical dissociation was that the processing time took less than 5 min per sample, as opposed to tens of minutes or even several hours for enzymatic dissociation protocols. Moreover, enzymatic protocols typically require sample-dependent reagents that are often expensive and require special storage conditions, whereas mechanical dissociation can be performed in standard culture medium. Although different enzymatic protocols often enrich for specific cell types[56], we believe that the single-cell suspension from mechanical dissociation might be more representative of the actual populations in tissue, and that it is therefore suitable for an unbiased examination of the cellular landscape. Fast dissociation also has the potential to preserve biochemical and biophysical properties of cells in a state near to in situ; these properties are likely to deteriorate with longer processing times in other approaches. Owing to the speed of mechanical dissociation, cells might undergo less proteomic or transcriptional changes, which are known to happen during enzymatic processing[34,57–60]. Further comparative and molecular studies are necessary to assess these assumptions.

In future, investigations on larger patient cohorts will allow to exploit machine learning for diagnostic or prognostic decision making. Artificial intelligence is already aiding pathologists in inspecting histological whole-slide images, diagnosing cancer or classifying tumours[61–63]. The large datasets obtained by RT–FDC analysis, composed of thousands of cell images and multi-dimensional information, lend themselves for such artificial intelligence approaches. In this study, we focused on parameters calculated from images in real time, but additional physical phenotype parameters can be calculated post-acquisition, and used as further inputs for machine learning, such as shape or texture features. Future work will also focus on the correlation between the physical phenotype data and tumour malignancy scoring, metastatic potential and survival rate.

Finally, an important aspect of the method is that the physical phenotype of cells can be used to identify cell populations in tissue, either in a fully label-free manner or synergistically with molecular markers, enhancing the fluorescence measurements. Furthermore, owing to the sorting modality recently added to RT–FDC[36], a specific population of cells can be isolated according to parameters calculated from images in real time or using trained neural networks[64,65]. This could be employed for enrichment of uncharacterized cell populations in tissue for downstream omics analysis or even for regenerative medicine purposes, such as for the label-free isolation of tissue-derived stem cells.

Overall, our findings show that the physical phenotyping of cells via RT–FDC after enzyme-free mechanical tissue dissociation is a quick and simple method that can be used to diagnose pathological states in tissue biopsies. In particular, it may provide a rapid and unbiased prediction of disease state in inflammatory conditions and in malignancy.

## Methods
### Animal experiments
All animal experiments were conducted in collaboration with the Department of Internal Medicine 1, University Hospital Erlangen, in compliance with all institutional and ethical guidelines, and covered by appropriate animal licences (Tierversuchsantrag no. 55.2.2- 2532-2-1032/55.2.2- 2532-2-473). Animal studies were conducted in a gender- and age-matched manner using littermates for each experiment. Both male and female animals were used. All mice were kept under specific pathogen-free conditions. Mice were routinely screened for pathogens according to Federation of European Laboratory Animal Science Associations guidelines. Mice were housed in 12 h light–dark cycle, at 20–23 °C and 40–60% humidity. Experiments were performed in accordance to the guidelines of the Institutional Animal Care and Use Committee of the State Government of Middle Franconia. Animals were killed by cervical dislocation and organs were surgically removed. For comparison of enzymatic and TG processing, female and male C57BL/6J mice were used, age 8–19 weeks. Lung and liver

tissues perfusion preceded the enzymatically dissociation process. For mechanical dissociation using a TG, organs were washed thoroughly with phosphate buffer solution (PBS) before being placed in Dulbecco's modified Eagle medium (DMEM) supplemented with 2% foetal bovine serum (FBS) and placed on ice until further processing. Enzymatic protocols were obtained from literature and are summarized in Supplementary Table 1. For both the enzymatic protocols and the TG, the weight of the tissue used was recorded. At the end of the dissociation procedure the total cell yield was counted using a LUNA cell counter.

### Adoptive lymphocyte transfer colitis
Immunodeficient Rag1$^{-/-}$ mice received 1 million CD4+ CD25− T cells via intraperitoneal injection. Mononuclear cells were isolated from the spleen of C57/BL6 donor mice and purified using magnetic-activated cell sorting technology, before being injected into immunodeficient mice as previously described[66,67]. Animals were killed 3 weeks after cell transfer and the colon tissue was processed as described in the 'Tissue dissociation and single-cell preparation section'.

### Spontaneous tumour model
To generate a specific deletion of an intestinal epithelial cell-specific protein with a key role for epithelial integrity, mice carrying LoxP-Cre flanked for the specific protein were cross-bred with VillinCre mice. Spontaneous tumorigenesis was observed in colon with 100% penetrance.

### Human tissue preparation
Surgically resected human biopsy samples (obtained from the Pathology Institute, Erlangen) from tumour or healthy tissue were immediately placed in Advanced DMEM medium supplemented with 10% FBS, 1% GlutMAX, 1% HEPES and 1% penicillin/streptomycin and stored at 4 °C, processed immediately or frozen in liquid nitrogen for later use. Matched pairs of samples were analysed, with two samples derived from each patient: a tumour sample and a control sample originating from healthy tissue surrounding the tumour.

The biopsy samples were not collected specifically for this research study but were part of the standard practices of patient care. Informed consent was obtained from patients providing samples and all experiments were carried out in accordance with the declaration of Helsinki. The protocol for obtaining human biopsy samples for this study was approved by ethic votes of the University Hospital of the Friedrich-Alexander University Erlangen-Nürnberg (24 January 2005, 18 January 2012; Institutional Review Board of the University Hospital of the Friedrich-Alexander University Erlangen-Nürnberg approval number: Re.-No. 4607).

Supplementary Tables 5–10 present the population characteristics and pathological information for all analysed human samples. Stromal tumour infiltrating lymphocytes and stroma content of tumours was scored by pathologists as described previously[68].

### Tissue dissociation and single-cell preparation
Tissue dissociation using a TG (Fast Forward Discoveries GmbH) was performed, as described in refs. 34,35. Briefly, the tissue sample was cut into small pieces of about 1–2 mm and placed into the rotor unit of the TG with 800 µl of DMEM supplemented with 2% FBS. The rotor unit was positioned in the lid of a 50 ml Falcon tube; the stator insert with a 100 µm cell strainer was placed on top of the rotor unit. A 50 ml Falcon tube was placed on the lid, screwed and positioned on the TG device (Fig. 1). The grinding process parameters for each tissue type are summarized in Supplementary Table 2. TG protocols were provided by the manufacturer with some minor modifications[34,35]. Following the grinding procedure, the Falcon tube was inverted onto a rack, opened and the cell strainer washed with 5 ml of DMEM, 2% FBS. The flow through was transferred into a 15 ml Falcon tube and centrifuged for 8 min at 300g. Subsequently, the supernatant was aspirated, and the cell pellet

washed with 2 ml of PBS, 2% FBS, passed through a flow cytometry round bottom tube with a cell strainer cap and centrifuge at 300*g* for 5 min. The cell pellet was resuspended in the high-viscosity measurement buffer prepared using 0.6% (wt/wt) methyl cellulose (4,000 cPs; Alfa Aesar) diluted in PBS without calcium and magnesium, adjusted to an osmolality of 270–290 mOsm kg$^{-1}$ and pH 7.4. The viscosity of the buffer was adjusted to $(25 \pm 0.5)$ mPa s$^{-1}$ at 24 °C using a viscometer (HAAKE Falling Ball Viscometer Type C, Thermo Fisher Scientific).

### RT–FDC

RT–FDC measurements were performed as previously described[13,14], using an AcCellerator instrument (Zellmechanik Dresden GmbH). The cell suspension was drawn into a 1 ml Luer-Lok syringe (BD Biosciences) attached to a syringe pump and connected by PEEK-tubing (IDEX Health & Science LLC) to a microfluidic chip made of polydimethylsiloxane bonded on a cover glass. A second syringe filled with pure measurement buffer was attached to the chip and used to hydrodynamically focus the cells inside the constriction channel. The microfluidic chip consisted of a sample inlet, a sheath inlet and an outlet connected by a central channel constriction of a $20 \times 20$, $30 \times 30$ or $40 \times 40$ μm square cross-section and a length of 300 μm. The corresponding total flow rates used were: 0.06 μl s for 20 μm$^{-1}$ channels, 0.12 μl s$^{-1}$ for 30 μm channels and 0.2 μl s$^{-1}$ for 40 μm channels. The sheath to sample flow ratio was 3:1. The chip was mounted on the stage of an inverted high-speed microscope equipped with a high-speed complementary metal-oxide semiconductor camera. The laser power for each fluorophore was adjusted accordingly, based on single stain controls and an unstained sample. An image of every cell was captured in a region of interest of $250 \times 80$ pixels at a frame rate of 2,000 fps. Morphological, mechanical and fluorescence parameters were acquired in real time. The fluorescence threshold for each antibody was adjusted according to an unstained sample of cells obtained from the same tissue. Supplementary Table 3 lists the features acquired in real time and during post-processing analysis; described in detail in previous publications[69,70]. Data were acquired using ShapeIn software (ShapeIn2; Zellmechanik Dresden GmbH).

### Fluorescence labelling

Where necessary, single-cell suspensions were incubated for 20 min at room temperature with 200 μl of corresponding antibodies (for antibodies dilution, see Supplementary Table 4) diluted in PBS supplemented with 0.5% bovine serum albumin (Sigma-Aldrich) and Fc receptor blocking reagent of corresponding species (Miltenyi Biotec, human: 130-059-901; mouse: 130-092-575). The antibodies were washed by adding 1 ml of PBS and 2% FBS, and centrifuged for 500*g* for 5 min. The final cell preparation was then resuspended in the measurement buffer before loading onto the microfluidic chip for RT–FDC analysis. For frozen biopsy samples, the tissue was placed in pre-warmed DMEM, supplemented with 10% FBS for 10 min and allowed to thaw before processing as described above.

### Data analysis

RT–FDC data were analysed using public packages in Python 3.7. Dclab 0.32.3 library was used for the initial loading, pre-processing and filtering of the data[71]. To remove images of debris, damaged cells and red blood cells, we applied gates for minimum cross-sectional area (20 μm$^2$), area ratio (1:1.1) and aspect ratio (1:2). Small cells <60 μm$^2$ were identified by additionally gating for area ratio 1:1.05 and aspect ratio 1:2. Any events outside of these gates and all events <25 μm$^2$ were considered as debris. In the scatter plots of RT–FDC parameters, colour coding is according to kernel density estimates normalized between 0 and 1.

Statistical analysis was done using the SciPy 1.3.0 package. The Wilcoxon signed rank test was used to assess paired samples (murine healthy versus tumour samples and human fresh colon tissue samples versus frozen samples). A Mann–Whitney *U* test was applied on transfer

colitis data. In graphs, *P* values are represented by * *P* < 0.05, ** *P* < 0.01 and *** *P* < 0.001. Effect sizes were calculated as $r = |z|/\sqrt{N}$, where *z* is the *z* statistic of the test and *N* is the number of samples. Effect sizes were judged according to Cohen criteria as follows: 0.1–0.3 small effect, 0.3–0.5 moderate effect and >0.5 large effect. Pearson's correlation was performed to judge the correlation between cell deformation and the number of CD45+ cells and the correlation between cell size and area of murine healthy and tumour samples.

The Scikit learn 0.23.2 package was used for further data processing and analysis[72]. Parameters obtained from RT–FDC were transformed by scaling each feature to the range between 0 and 1. PCA was used for linear dimensionality reduction, using singular value decomposition of the data to project onto a two-dimensional space (PC1 versus PC2). Logistic regression was used for the classification of healthy versus tumour samples.

### Reporting summary

Further information on research design is available in the Nature Portfolio Reporting Summary linked to this article.

### Data availability

The RT–FDC datasets generated and analysed for Figs. 2–5 and Extended Data Figs. 3–5 are available on the Deformability Cytometry Open Repository (https://dcor.mpl.mpg.de/organization/soteriou-kubankova)[73]. Individual identifiers for each dataset are provided in Supplementary Table 11. Source data for Extended Data Fig. 1 are also provided with this paper. Source data are provided with this paper.

### Code availability

The Python code for the processing and visualization of RT–FDC data is available at https://github.com/marketakub/physical_phenotyping_tissues.

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

## Acknowledgements

We thank R. Grützmann and K. Bende (Universitätsklinikum Erlangen) for kindly assisting us with biopsy samples. We also thank J. Kayser and M. Urbanska for a critical review of the paper. We acknowledge financial support from the DFG (SFB/TRR241 'IBDome' to R.L.P., M.W., R.A. and M.N.; FOR2438 to M.N; DI 1537/20-1, DI 1537/22-1, SFB CRC1181 and SFB TR221 to J.D.), the IZKF Erlangen (research grant A79 to J.D. and the clinician scientist programme step 2 to M.E.), and Max Planck Society core funding (to J.G.).

## Author contributions

M.K., D.S. and J.G. conceived the study and managed the project progress. D.S. and M.K. coordinated the experiments and analysis. D.S. developed experimental protocols and performed experiments together with C.S. Data analysis and visualization was performed by M.K. Other authors assisted with the experiments and participated in critical discussions. M.K. and D.S. wrote the paper, with all authors providing feedback.

## Funding

## Competing interests

S.S., J.L. and J.G. are co-founders of the company Fast Forward Discoveries GmbH, which commercializes the TG technology. D.S., M.K. and J.G. are named inventors on a patent application for the combination of TG and RT–DC for solid biopsy diagnosis. M.K. and J.G. are co-founders of the company Rivercyte GmbH, which commercializes diagnostic applications of deformability cytometry. The other authors declare no competing interests.

## Additional information

**Extended data** is available for this paper at https://doi.org/10.1038/s41551-023-01015-3.

**Correspondence and requests for materials** should be addressed to Jochen Guck.

**a**

Cell viability assessed with propidium iodide

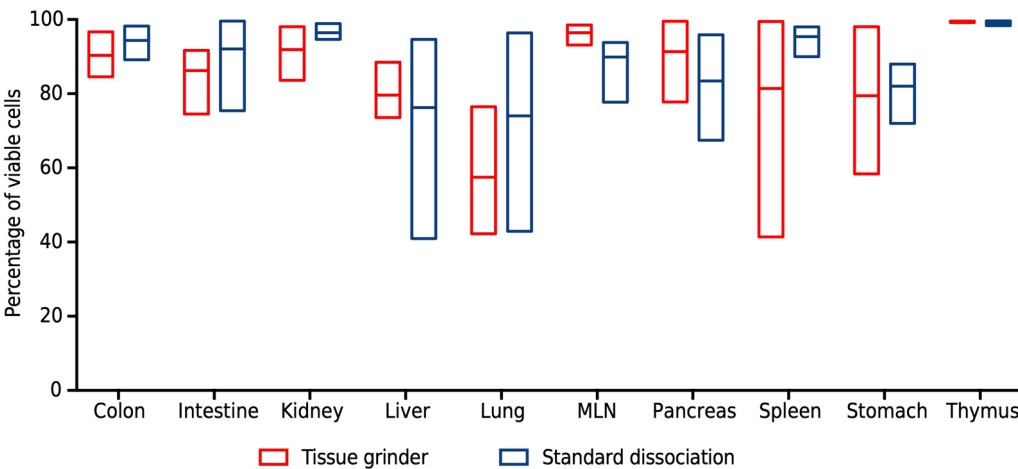

**b**

Cell viability assessed with trypan blue

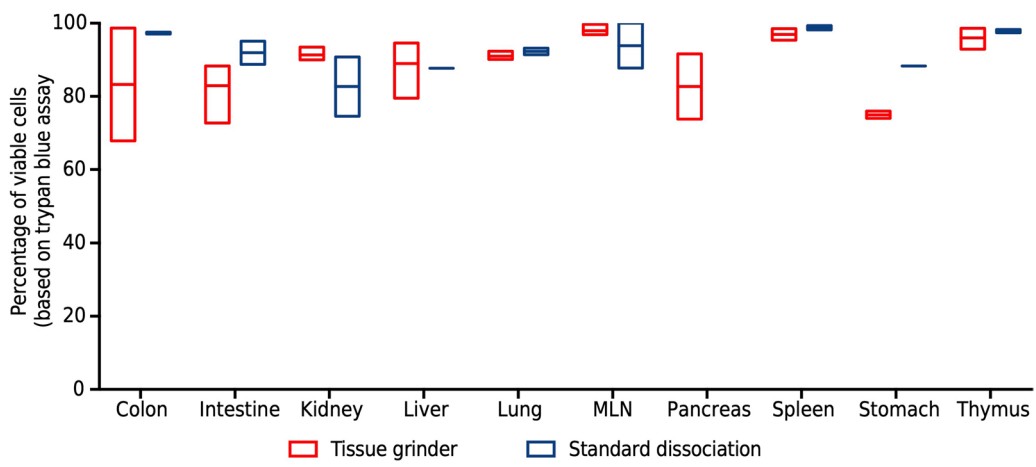

**c**

Total number of cells per mg of tissue

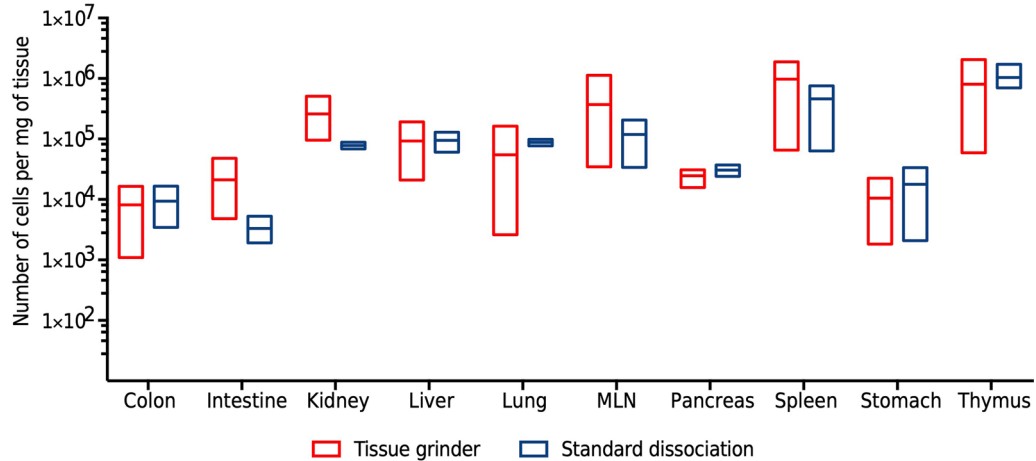

**Extended Data Fig. 1 | See next page for caption.**

**Extended Data Fig. 1 | Comparison of cell viability and cell yield of mechanical vs standard dissociation of different murine tissues. a,b**, Percentage of viable cells for different organs dissociated using a tissue grinder (TG; marked in red) or standard dissociation (marked in blue). Cell viability was assessed using (**a**) propidium iodide and RT-FDC and (**b**) trypan blue exclusion assay. **c**, Number of cells (obtained using a cell counter device) per mg of tissue processed. Lung, liver, kidney, pancreas and stomach processed with enzymatic dissociation were not weighted prior to the experiments. The line represents the mean and the box extends from minimum to maximum values. TG data kidney and spleen: n = 5, all other organs n = 5; for enzymatic dissociation kidney, pancreas, stomach: n = 3; all other organs n = 4; biologically independent repeats performed as independent experiments).

**a**

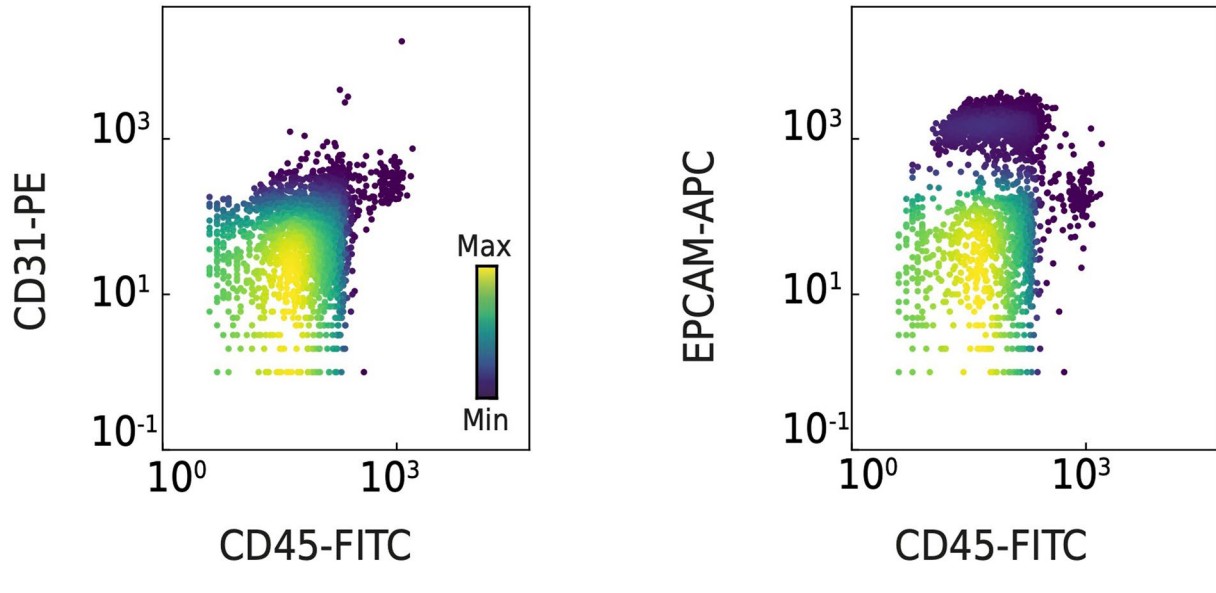

**b**

EpCAM-FITC      CD31-PE      CD45-APC

**Extended Data Fig. 2 | See next page for caption.**

**Extended Data Fig. 2 | Detection of fluorescent cell surface markers using RT-FDC. a**, Representative scatter plots of fluorescence intensities in three different detection channels of RT-FDC, showing the possibility of using fluorescent cell surface markers to characterize the cells. Plots show expression of an endothelial marker (CD31-PE) vs leucocyte marker (CD45-FITC); and an epithelial marker (EpCAM-APC) vs CD45-FITC. **b**, Representative images of cells and their corresponding fluorescence traces; the temporal shape of the fluorescence peak corresponds to the subcellular localization of the fluorophore. Top left to right: cell negative for all markers, a cell positive for CD45; bottom left to right: cell positive for EpCAM only and cell positive for CD31 only.

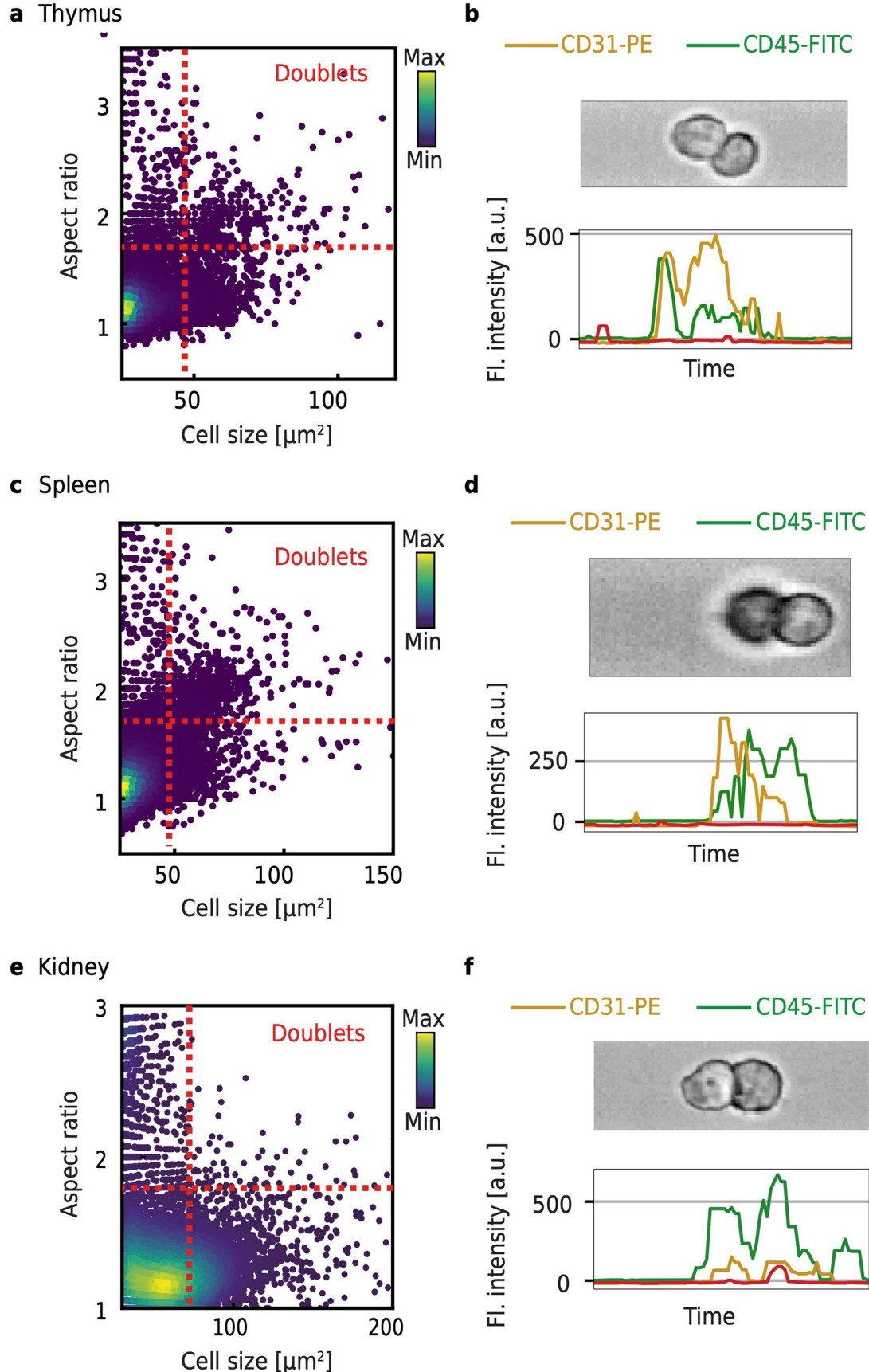

**Extended Data Fig. 3 | Detection of cell doublets using RT-FDC.** Representative scatter plots of aspect ratio vs cell size of cells isolated from murine **a**, thymus, **c**, spleen and **e**, kidney showing the gating strategy for identifying cell doublets. Cell doublets identified in **b**, thymus and **d**, spleen and **f**, kidney with corresponding fluorescence traces (**b**,**d**), showing a leucocyte (CD45) attached to an endothelial cell (CD31), or (**f**) the interaction of two leukocytes.

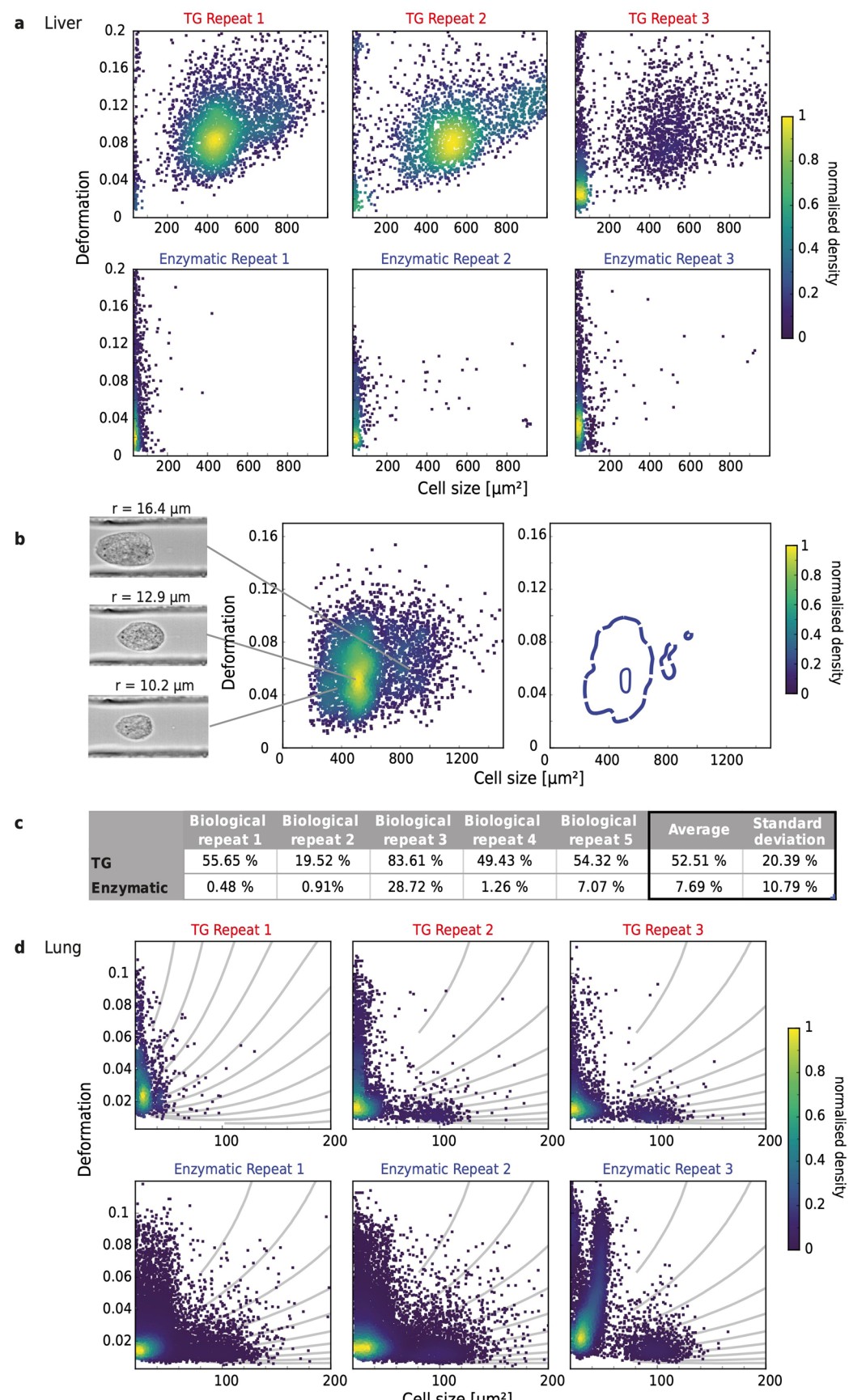

**Extended Data Fig. 4 | See next page for caption.**

**Extended Data Fig. 4 | Physical phenotype characterisation of cells isolated mechanically and enzymatically from murine lung and liver. a**, Scatter plots of deformation vs cell size for cells isolated from mouse liver tissue using a tissue grinder (TG) or enzymatic dissociation, showing the enrichment of hepatocytes following mechanical dissociation for 3 independent biological repeats. **b**, Scatter plot of deformation vs cell size showing 3 clusters of cells that correspond to hepatocytes of different sizes; with the corresponding kernel density estimate (KDE) plot and representative images (r = radius of cells). **c**, Percentage of hepatocytes to the total number of liver cells, as detected by RT-DC for five independent biological repeats. **d**, Scatter plots of deformation vs cell size for cells isolated from mouse lung tissue using a tissue grinder or enzymatic dissociation for 3 independent biological repeats.

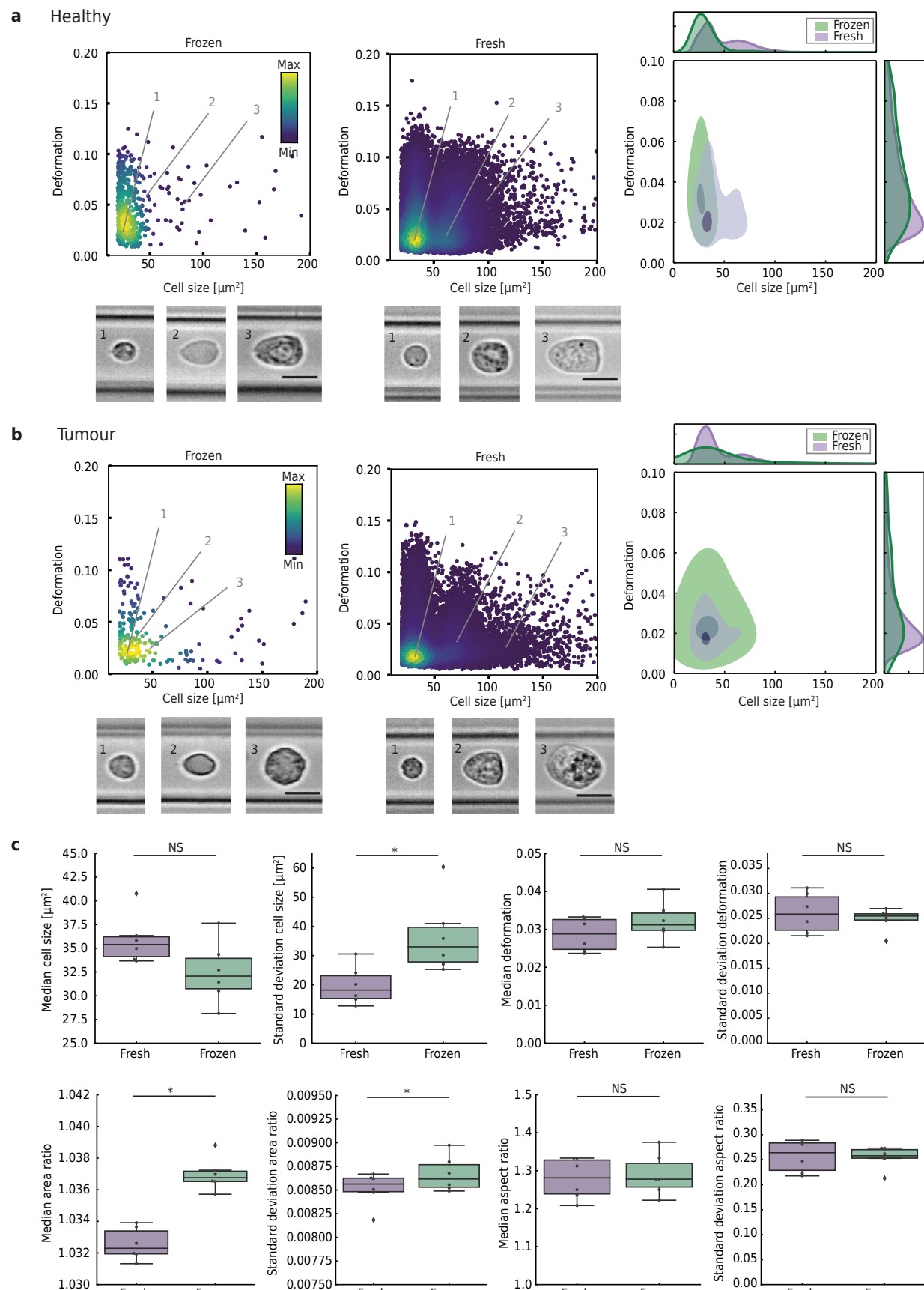

**Extended Data Fig. 5 | See next page for caption.**

**Extended Data Fig. 5 | Comparison of physical phenotype parameters of cells from frozen and fresh human biopsy samples.** Cell size vs deformation scatter plots of single cells extracted from either fresh (purple) or frozen (green) colon biopsy samples; **a**, healthy sample; **b**, tumour sample; including 3 representative cell images for each plot with a scale bar = 10 μm². The kernel density estimate (KDE) plots on the right correspond to the scatter plots on the left; the histograms show the distributions of cell size and deformation. **c**, Medians and standard deviations of cell size, deformation, area ratio and aspect ratio of fresh ($N = 6$) and corresponding frozen ($N = 6$) samples. Boxes extend from the 25th to the 75th percentile with a line at the median; whiskers span 1.5x the interquartile range. Statistical comparisons were performed using a two-sided Wilcoxon signed rank test, standard deviation cell size (*$p = 0.0277$, $r = 0.64$), median area ratio (*$p = 0.0277$, $r = 0.64$) and standard deviation area ratio (*$p = 0.0277$, $r = 0.64$); $r$: effect size; NS: non-significant.

# Reporting Summary

## Statistics

For all statistical analyses, confirm that the following items are present in the figure legend, table legend, main text, or Methods section.

| n/a | Confirmed | |
|---|---|---|
| ☐ | ☒ | The exact sample size (*n*) for each experimental group/condition, given as a discrete number and unit of measurement |
| ☒ | ☐ | A statement on whether measurements were taken from distinct samples or whether the same sample was measured repeatedly |
| ☐ | ☒ | The statistical test(s) used AND whether they are one- or two-sided *Only common tests should be described solely by name; describe more complex techniques in the Methods section.* |
| ☒ | ☐ | A description of all covariates tested |
| ☒ | ☐ | A description of any assumptions or corrections, such as tests of normality and adjustment for multiple comparisons |
| ☐ | ☒ | A full description of the statistical parameters including central tendency (e.g. means) or other basic estimates (e.g. regression coefficient) AND variation (e.g. standard deviation) or associated estimates of uncertainty (e.g. confidence intervals) |
| ☐ | ☒ | For null hypothesis testing, the test statistic (e.g. *F*, *t*, *r*) with confidence intervals, effect sizes, degrees of freedom and *P* value noted *Give P values as exact values whenever suitable.* |
| ☒ | ☐ | For Bayesian analysis, information on the choice of priors and Markov chain Monte Carlo settings |
| ☒ | ☐ | For hierarchical and complex designs, identification of the appropriate level for tests and full reporting of outcomes |
| ☐ | ☒ | Estimates of effect sizes (e.g. Cohen's *d*, Pearson's *r*), indicating how they were calculated |

*Our web collection on statistics for biologists contains articles on many of the points above.*

## Software and code

Policy information about availability of computer code

| | |
|---|---|
| Data collection | We used ShapeIn (ShapeIn2; commercially available from Zellmechanik Dresden GmbH) for the acquisition of the RT-FDC data. |
| Data analysis | Python 3.7 was used. The Python library dclab is open-source and available on github (https://github.com/ZELLMECHANIK-DRESDEN/dclab); package version 0.32.3 was used for data analysis. We performed statistical analyses with the SciPy 1.3.0 package. Additional data analysis was performed with Scikit learn 0.23.2 package (https://scikit-learn.org/stable/). The Python code for the processing and visualisation of RT-FDC data is available at https://github.com/marketakub/physical_phenotyping_tissues. |

For manuscripts utilizing custom algorithms or software that are central to the research but not yet described in published literature, software must be made available to editors and reviewers. We strongly encourage code deposition in a community repository (e.g. GitHub). See the Nature Portfolio guidelines for submitting code & software for further information.

## Data

Policy information about availability of data

All manuscripts must include a data availability statement. This statement should provide the following information, where applicable:

- Accession codes, unique identifiers, or web links for publicly available datasets
- A description of any restrictions on data availability
- For clinical datasets or third party data, please ensure that the statement adheres to our policy

The RT-FDC datasets generated and analysed for Figs. 2–5 and Extended Data Figs. 3–5 are available on the Deformability Cytometry Open Repository (https://dcor.mpl.mpg.de/organization/soteriou-kubankova)73. Individual identifiers for each dataset are provided in Supplementary Table 11. Source data for Extended Data Fig. 1 are also provided with this paper.

## Human research participants

Policy information about studies involving human research participants and Sex and Gender in Research.

| Reporting on sex and gender | Sex and gender were not considered in the study design nor in the data analysis. Our working assumption was that there are no sex-related or gender-related differences in colon tumours; hence, sex and gender were not considered as relevant parameters in our analysis. |
|---|---|
| Population characteristics | Surgically resected human biopsy samples from male and female patients of age range 57–88 were obtained from the Pathology Institute, Erlangen. All samples used were obtained from leftover biopsy samples. Therefore, this work did not interfere with standard practices of care or with sample-collection procedures. We provide the main population characteristics, including age, gender, diagnosis, localization of biopsy, histology, pT and pN stage, tumour grade, resection status, and the characteristics of the stroma and invasion state, in the Supplementary Information. |
| Recruitment | No active recruitment was needed. We used leftover biopsy samples that were obtained (subject to availability) from the Pathology Institute, Erlangen, following patient surgery. The biopsy samples were not collected specifically for this research study but were part of the standard practices of patient care. Informed consent was obtained from the patients providing samples. The participants were not compensated. All experiments were carried out in accordance with the declaration of Helsinki. |
| Ethics oversight | The study is covered by ethic votes of the University Hospital of the Friedrich-Alexander University Erlangen-Nurnberg (24.01.2005, 18.01.2012). The Institutional Review Board of the University Hospital of the Friedrich-Alexander University Erlangen-Nürnberg approved the study (ID: Re.-No. 4607). |

Note that full information on the approval of the study protocol must also be provided in the manuscript.

# Field-specific reporting

Please select the one below that is the best fit for your research. If you are not sure, read the appropriate sections before making your selection.

☒ Life sciences   ☐ Behavioural & social sciences   ☐ Ecological, evolutionary & environmental sciences

For a reference copy of the document with all sections, see nature.com/documents/nr-reporting-summary-flat.pdf

# Life sciences study design

All studies must disclose on these points even when the disclosure is negative.

| Sample size | No statistical method was used to predetermine sample sizes. The sample sizes used in the study are in accordance with past experience and with standard sizes used in real-time deformability-cytometry measurements, which are of the order of several hundreds of cells per measurement per condition. The number of mice used in the animal studies were estimated according to the Resource Equation method [Charan et al. (2013), J Pharmacol Pharmacother]. For human biopsies, sample sizes were determined by the number of patient samples available. |
|---|---|
| Data exclusions | For the analysis of real-time deformability-cytometry data, we excluded debris by filtering out events smaller than 20 μm^2 in cross-sectional area, a procedure commonly used in real-time deformability cytometry. An additional area-ratio filter of 1.0–1.1 was used to ensure that only events with correctly fitted contours are used in the data analysis, as previously established by ZellMechanik Dresden GMBH, Dresden, Germany. |
| Replication | All of the experiments presented in the manuscript were repeated more than once, as indicated in the figure legend. All experiments were included in the analysis. |
| Randomization | Randomization was not relevant for the study, because no treatment group was involved. |

| Blinding | For blind data analysis, samples were numerically tagged. The investigator running the analysis was unaware of which sample corresponded to tumour or healthy tissue. Only the investigators acquiring the data were aware of the status of the biopsy. Owing to the appearance of the biopsy samples, it was in some cases clear to the investigator performing the experiments which sample corresponded to tumour or healthy tissue. It was therefore impossible to perform blind data acquisition. |
|---|---|

# Reporting for specific materials, systems and methods

We require information from authors about some types of materials, experimental systems and methods used in many studies. Here, indicate whether each material, system or method listed is relevant to your study. If you are not sure if a list item applies to your research, read the appropriate section before selecting a response.

### Materials & experimental systems

| n/a | Involved in the study |
|---|---|
| ☐ | ☒ Antibodies |
| ☒ | ☐ Eukaryotic cell lines |
| ☒ | ☐ Palaeontology and archaeology |
| ☐ | ☒ Animals and other organisms |
| ☒ | ☐ Clinical data |
| ☒ | ☐ Dual use research of concern |

### Methods

| n/a | Involved in the study |
|---|---|
| ☒ | ☐ ChIP-seq |
| ☒ | ☐ Flow cytometry |
| ☒ | ☐ MRI-based neuroimaging |

## Antibodies

| Antibodies used | anti-human: CD326 (EpCAM) Alexa Fluor 488 (1:100, Clone:9C4, Ref :324210; BioLegend, CA, USA); CD31(PECAM-1) APC (5µl/reaction, Clone: WM59, Ref: 303115; BioLegend, CA, USA); CD45-PE (1:500. Clone HI30, Ref: 304008; BioLegend, CA, USA); CD326 (EpCAM) FITC (1:500, Clone:9C4, Ref: 324203; BioLegend, CA, USA); CD45 Alexa Fluor 700 (5µl/reaction, Clone: HI30, Ref: 304024; BioLegend, CA, USA). Anti-mouse: CD45 Alexa Fluor® 700 (1:1000, Clone: 30-F11, Ref: 103128; BioLegend, CA, USA); CD326 (EpCAM) Alexa Fluor 488 (1:200, Clone: G8.8, Ref: 118210; BioLegend, CA, USA); CD45-FITC (1:800. Clone: 30-F11, Ref: 11-0451-82; Thermo Fischer Scientific, MA, USA); CD326 (EpCAM) APC (1:500, Clone: G8.8, Ref: 17-5791-82; Thermo Fischer Scientific, MA, USA); CD31 (PECAM) PE (1:250, Clone:390, Ref: 12-0311-82; Thermo Fischer Scientific, MA, USA). |
|---|---|
| Validation | Antibodies were validated by the manufacturers and were used according to manufacturers' recommended dilutions. Anti-human: CD326 (EpCAM) Alexa Fluor 488: https://www.biolegend.com/en-ie/search-results/alexa-fluor-488-anti-human-cd326-epcam-antibody-3759 CD31(PECAM-1) APC: https://www.biolegend.com/en-ie/products/apc-anti-human-cd31-antibody-6123 CD45 PE: https://www.biolegend.com/en-ie/products/pe-anti-human-cd45-antibody-708 CD326 (EpCAM) FITC: https://www.biolegend.com/en-ie/products/fitc-anti-human-cd326-epcam-antibody-3756 CD45 Alexa Fluor 700: https://www.biolegend.com/en-ie/products/alexa-fluor-700-anti-human-cd45-antibody-3401<br><br>Anti-mouse: CD45 Alexa Fluor 700 : https://www.biolegend.com/en-ie/products/alexa-fluor-700-anti-mouse-cd45-antibody-3407 CD326 (EpCAM) Alexa Fluor 488 :https://www.biolegend.com/en-ie/products/alexa-fluor-488-anti-mouse-cd326-ep-cam-antibody-4972 CD45 FITC: https://www.thermofisher.com/antibody/product/CD45-Antibody-clone-30-F11-Monoclonal/11-0451-82 CD326 (EpCAM) APC: https://www.thermofisher.com/antibody/product/CD326-EpCAM-Antibody-clone-G8-8-Monoclonal/17-5791-82 CD31 (PECAM-1) PE: https://www.thermofisher.com/antibody/product/CD31-PECAM-1-Antibody-clone-390-Monoclonal/12-0311-82 |

## Animals and other research organisms

Policy information about studies involving animals; ARRIVE guidelines recommended for reporting animal research, and Sex and Gender in Research

| Laboratory animals | For the comparison of enzymatic and tissue grinding: C57BL/6J females and males, age 8–19 weeks; For the transfer colitis model: Mus musculus, Rag1-/-, female, 8–12 weeks at the start of the experiment. For the tumour model: Mus musculus, female and male mice, 30–40 weeks. Animals were housed in 12-hour light–dark cycle, at 20–23°C and 40–60 % humidity. |
|---|---|
| Wild animals | The study did not involve wild animals. |
| Reporting on sex | Male (N=3) and female (N=4) animals were used for the comparison of enzymatic dissociation and tissue grinding. For each tissue extracted, half were used for enzymatic dissociation and half for tissue grinding.<br><br>Only female animals were used for the transfer colitis experiments (N=14), as described in the original publication (DOI: 10.1093/intimm/5.11.1461) |

Both male (N=7) and female (N=9) animals were used for the tumour model.

No sex-based analysis was performed because this was beyond the scope of this study.

Field-collected samples

The study did not involve samples collected from the field.

Ethics oversight

The experiments were performed in accordance to the guidelines of the Institutional Animal Care and Use Committee of the State Government of Middle Franconia.
Transfer colitis: TVA: 55.2.2- 2532-2-473; Government of Lower Franconia
Tumour model: TVA: 55.2.2- 2532-2-1032; Government of Lower Franconia

Note that full information on the approval of the study protocol must also be provided in the manuscript.

