## [Peer Review File · Nature Biomedical Engineering]

Rapid single-cell physical phenotyping of mechanically dissociated tissue biopsies

Corresponding author: Jochen Guck

Editorial note

This document includes relevant written communications between the manuscript's corresponding author and the editor and reviewers of the manuscript during peer review. It includes decision letters relaying any editorial points and peer-review reports, and the authors' replies to these (under 'Rebuttal' headings). The editorial decisions are signed by the manuscript's handling editor, yet the editorial team and ultimately the journal's Chief Editor share responsibility for all decisions.

Any relevant documents attached to the decision letters are referred to as **Appendix #**, and can be found appended to this document. Any information deemed confidential has been redacted or removed. Earlier versions of the manuscript are not published, yet the originally submitted version may be available as a preprint. Because of editorial edits and changes during peer review, the published title of the paper and the title mentioned in below correspondence may differ.

Correspondence

Fri 14 Jan 2022

Decision on Article nBME-21-2659

Dear Prof Guck,

Thank you again for submitting to *Nature Biomedical Engineering* your manuscript, "Single-cell physical phenotyping of mechanically dissociated tissue biopsies for fast diagnostic assessment". The manuscript has been seen by three experts, whose reports you will find at the end of this message. You will see that the reviewers appreciate the work, and that they raise a number of technical criticisms that we hope you will be able to address. In particular, we would expect that a revised version of the manuscript provides:

- * Extended evidence of the viability of the cells dissociated via the tissue grinder, with respect to traditional enzymatic methods.
- * Evidence of the classification performance of physical phenotyping in tissues that are more representative of clinical scenarios, as per the comments of all reviewers.
- * Enhanced characterization of the performance of cell sorting via physical phenotyping, in terms of the range of detectable phenotypes, robustness and reproducibility.
- * Thorough methodological details.

When you are ready to resubmit your manuscript, please upload the revised files, a point-by-point rebuttal to the comments from all reviewers, the reporting summary, and a cover letter that explains the main improvements included in the revision and responds to any points highlighted in this decision.

Please follow the following recommendations:

- * Clearly highlight any amendments to the text and figures to help the reviewers and editors find andunderstand the changes (yet keep in mind that excessive marking can hinder readability).

* If you and your co-authors disagree with a criticism, provide the arguments to the reviewer (optionally, indicate the relevant points in the cover letter).

* If a criticism or suggestion is not addressed, please indicate so in the rebuttal to the reviewer comments and explain the reason(s).

* Consider including responses to any criticisms raised by more than one reviewer at the beginning of the rebuttal, in a section addressed to all reviewers.

* The rebuttal should include the reviewer comments in point-by-point format (please note that we provide all reviewers will the reports as they appear at the end of this message).

* Provide the rebuttal to the reviewer comments and the cover letter as separate files.

We hope that you will be able to resubmit the manuscript within 15 weeks from the receipt of this message. If this is the case, you will be protected against potential scooping. Otherwise, we will be happy to consider a revised manuscript as long as the significance of the work is not compromised by work published elsewhere or accepted for publication at *Nature Biomedical Engineering*.

We hope that you will find the referee reports helpful when revising the work, which we look forward to receive. Please do not hesitate to contact me should you have any questions.

Best wishes,

Pep

Pep Pàmies
Chief Editor, Nature Biomedical Engineering

Reviewer #1 (Report for the authors (Required)):

The manuscript by Soteriou, et al., provides evidence that solid tissues can be quickly dissociated and analyzed using RT-FDC, and that this approach can be used for diagnostic purposes. Several examples are shown, including inflammatory bowel disease and the ability to distinguish between healthy and cancerous tissue, with the goal of establishing a label-free diagnostic pipeline based single-cell deformability, size and bright-field imaging properties. The RT-FDC achieves very high throughput when compared to typical single-cell biophysical measurements and the integration of deformability with bright-field imaging provides a unique way to assess cell state. A major strength of the manuscript is the inclusion of this approach with a rapid mechanical tissue dissociation to examine clinical specimens.

Although including multiple examples within the manuscript highlights the generality of the measurement platform, it comes at the expense of depth and establishing a compelling use case. Most central to the manuscript is distinguishing between healthy and cancer tissue. However, the scenarios examined involve tissues that are predominately cancer or predominately healthy, which could be identified macroscopically. The real challenge would be to find a small island of tumor (or small fraction of tumor cells) within an otherwise normal specimen or a similar clinical scenario of actual relevance. Thus, the clinical implications based on what is shown in the manuscript are not clear.

In Figure 2, the authors measure physical parameters of cells from murine liver, colon and kidney samples and then examine immunophenotypes for various sections of the scatter plots. In one example they show a cluster of cells with similar physical properties was mainly composed of EpCAM positive cells (Fig. 2a). The authors claim this shows 'a clean population of epithelial cells can be distinguished in a label-free manner, purely using image-derived physical parameters.' However examining arbitrary regions of scatter plots of

individual samples as is done throughout Figure 2 does not seem useful by itself. It would be more compelling if the authors could define a clear use case and then show how biophysical parameters leads to superior performance over immunophenotypic markers.

Major concerns:

1. The reproducibility of the work remains unclear. In several figures, such as figure 2, the authors display a single example sample, but the reproducibility of the conclusions across independent samples is unclear. Some figures, such as supplemental fig 1, have unacceptably low N numbers. In some figures, such as figure 3, the author report a higher N number, but whether the N refers to independent experiments or, for example, just technical replicates from the same patient, is not known. Considering the claims that the method should be adapted for routine clinical practice, including more blind-controlled experiments (as was done with a few samples in figure 5a) is essential.
2. Given the wide dimensions of disease spectrum, the authors should be more realistic in their discussion of the method potential as probably not all disease will have a strong signal to noise in deformability / imaging space. Without establishing the dynamic range of pathologic deformability vs normal range, it's difficult to assess whether the approach has broad vs. limited use.
3. The abstract has the strong claim that “[...] distinguish subpopulation of cells..[] without prior knowledge or the need for molecular marker”. The authors should elaborate more on the need to eventually or synergistically use molecular markers with the RT-FDC. For example, in Fig.2a it is not clear if the cluster of epithelial (EpCAM positive) cells (having a faintly lighter blue tone and artificially highlighted in green) could be necessarily picked out without relying on the molecular marker.
4. Related #3, the approach may be stronger when combining fluorescence readout with RT-FDC (Figs. 2b,c + Sup Fig. 3). The authors should consider amending their statement in the abstract by emphasizing enhancing the fluorescence measurement, rather than substituting it for single-cell measurements. Same for conclusion statement (lines 363-365).
5. When the authors describe the potential for sorting/studying cell-doublets (line 138-139), the value of the a qualitative measure of deformability (larger deviation from aspherical cells discussed in lines 332-334) is not clear. Can orientation be a big confounding factor in such a measurement?
6. The “Tissue Grinder – RT-FDC” data on lung tissue (Supp Fig. 4c) are not discussed within the text of paper. Also, it seems that enzymatic dissociation is biased towards lower cell sizes for liver, for lung it is not the case (this is evident in Run 2, while Run 1 has a sudden cut-off). Could the authors comment on why this happens? The current statement in the opening line of the paragraph (line 150) thus seems ambiguous, and the whole paragraph seems not well integrated with the rest of the text and figures.
7. In figure 2, the authors present a range of phenotypes that they can detect. The authors should address if these phenotypes reflect diverse cellular responses to the TG or RT-FDC, rather than phenotypes originally present in vivo. Furthermore, on line 157, the authors say that “using TG, the proportion of hepatocytes to total cells was between 40-80%”. The authors should justify this result and provide evidence for whether this is a purely a technical artifact or a biologically meaningful result.

Minor comments:

1. What is the high-viscosity medium in line 109? Are there special requirements that need attention? Please specify.
2. Line 128: In text it is stated that there are 6 clusters of cells in Fig. 2b while in the figure there are 7 clusters displayed.
3. It would help to remove various faint horizontal and vertical lines in Figure 3 and make sure that plots are consistently presented as closed or open boxes.
4. Figs. 3c,4c: include legend or text in caption to explain how the margins are drawn.
5. Generally Supp Fig. 4, if included, should have clear markings of the cell images as to where they belong in the RT-FDC graphs as it is done for example in Fig. 2b,c.
6. For consistency, in line 184, change “number” to “percentage” to be consistent with Fig. label on horizontal axes of Fig. 3d,e.

Reviewer #2 (Report for the authors (Required)):

The manuscript describes the application of high-throughput microfluidic single-cell physical phenotyping to solid tissue biopsies towards applications in diagnosis of diseased states in tissues. This is an important topic since physical phenotyping approaches can enable rapid and unbiased testing that allows, for example, measurement of malignancy during surgical procedures, rather than after histological sectioning, staining and analysis. Previous work in the field has mainly addressed cells naturally in suspension like in blood or other body fluids. The key breakthrough of this work is showing the ability to use a mechanical dissociation approach to obtain single cells compatible with the high-throughput microfluidic measurements. It appears that the approach to dissociation is also robust and repeatable enough to identify disease-specific signatures of cells across many samples. Several disease models are studied, including controlled animal models of colitis and malignancy, as well as human tissues (including previously frozen tissues). Overall, the work is well conducted and reported on an important and impactful topic. Some of the claims should be better substantiated with data analysis approaches. There should also be more clarity on the classification of the clinical disease states and what samples were used to train vs. validate the accuracy of the test.

Specific Comments:

Major:

Many claims throughout the manuscript point to the added value of parameters derived from cell deformation on disease diagnostic accuracy, however these parameters were not directly evaluated as useful compared to related parameters. For example, in the transfer colitis model, can the authors determine if the deformation parameters provided additional information beyond the amount or fraction of infiltrating immune cells which could be obtained using other processes (e.g. hematology analyzer following mechanical tissue dissociation). In the mouse tumor model, was there any correlation between deformation and size (and the statistics of these metrics) or were these parameters independent predictors of being a tumor sample? Understanding the correlation between the different parameters would be helpful to better elucidate what leads to the differential signal. Another way to test the importance of the deformation parameters is to train a model that does not use them and see how well that model performs. If you remove deformation parameters, can you still perform PCA to separate the samples for the human biopsy studies? How well does this perform?

For the fresh tumor biopsies it was unclear what the training / validation sets were for these experiments when reading the main text. If there were not separate validation data sets, this should be clearly indicated.

This reviewer was surprised that intact cells were observable from cryopreserved tissue sections, since there would be cell death and cell membranes compromised in this process. How does this affect the images/contrast? One would expect contrast would be decreased substantially given the ability of the suspending solution to transfer into the cell if the membrane is disrupted. Can the authors show images of cells observed from these various tissue sources to understand how the differences in cell images, initial circularity, and contrast, may affect the performance of the technique.

The authors state “The speed of the extraction presumably helps to preserve biochemical and biophysical phenotypes in conditions close to those in situ.”

It is unclear what evidence supports this statement. A main concern would be how mechanical shearing affects the physical properties of cells and whether some populations are damaged more or are more adhesive to other cells and stay in aggregates. Then these populations would be underrepresented in the final analysis process. Perhaps the authors could evaluate the cell type distributions in the tissue using staining approaches and compare to the cell types that make it through to single cells for analysis to determine if there is any bias / enrichment effect. I understand this information does not necessarily impact the classification results reported as it may be that differences in enrichment/depletion is part of the signal observed that leads to the differences between diseased and normal tissue.

Minor:

In Supplementary Fig. 1 – blue bars are labeled as enzymatic dissociation or mechanical dissociation. The caption indicated it should be enzymatic dissociation. The discrepancy should be corrected.

Reviewer #3 (Report for the authors (Required)):

This article by Despina Soteriou et al. entitled "Single-cell physical phenotyping of mechanically dissociated tissue biopsies for fast diagnostic assessment" presents a diagnostic pipeline for the rapid, label-free analysis of biopsy samples by sequentially assessing the physical phenotype of singularized, suspended cells in high-throughput. The authors demonstrate the potential of their method for inflammatory bowel disease diagnostics. Their results suggest that mechanically dissociating tissue biopsies can be used to accurately distinguish between healthy and tumor tissue in mouse and human biopsy samples. The authors claim that their method is quick and delivers results within 30 minutes, laying the groundwork for a fast and marker-free diagnostic pipeline to detect pathological changes in solid biopsies. The manuscript is well-written, and the figures clearly show workflows and results. However, I have several specific questions that I want the authors to address.

Specific comments

My main concern with this paper is the mechanical grinder method because it is harsh to the cells and should cause significant cell death. Cells are sticky and attach quite firmly to each other and therefore need a relatively large force to separate. Thus, enzymatic treatments are used so frequently. The authors must provide more robust evidence that the mechanical method is similar to or superior to the enzymatic methods. Without further evidence, this method lacks generality and can only be applied to a limited number of tissues.

In Supplementary Fig. 1, the authors compare the cell viability and cell yield between mechanical dissociation using a tissue grinder and enzymatic dissociation using conventional enzymes. The cell viability is assessed by propidium iodide and the cell yield is evaluated by RT-FDC. However, RT-FDC is not the standard method for determining cell viability and yield. The authors must perform similar experiments using FACS.

The pathological data of human samples used in this study is not presented. The authors need to show that their method works for both high-grade and low-grade cancers. In high-grade cancer, it is easier to detect differences in physical parameters, such as cell size; however, the differences in physical parameters are much more difficult to detect in low-grade cancer.

In Fig. 5, human cancer and control tissues are used. Were matched pairs used? The numbers suggest that control tissues were taken from the cancer patients, e.g., 13 frozen colon tumors vs. 13 frozen colon control, 11 fresh colon tumors vs. 11 fresh colon control, and 7 fresh lung tumors vs. 7 fresh lung control. The authors must show how well their method can detect differences in cancer and control tissue from the same patient. This is essential since this is how the method is suggested to be used.

In Fig. 2a (liver), the authors show that a subpopulation of cells forms a cluster at a specific cell size and brightness and that these cells are uniform in the enrichment of CD31, CD45, and EpCAM. Why was this cluster chosen? How about other subpopulations from the same scatter plot? Do they have different enrichments of CD31, CD45, and EpCAM?

The manuscript shows that RT-FDC can detect cell doublets from a cell suspension from murine thymus and spleen. There are many lymphocytes in the thymus and spleen, and they are relatively uniform in their shape, so cell doublets can easily be detected using RT-FDC. The authors should also show that cell doublets can be detected in other solid organs, such as the colon and kidney.

In Supplementary Fig. 6, the authors show that a high percentage of CD45 positive cells are smaller in size than $60 \mu\text{m}^2$. These smaller cells are excluded from the analysis as these cells are mainly "immune cells and small debris". Please back up this exclusion with experimental data or at least a reference. Of all excluded cells ($<60 \mu\text{m}^2$), what percentage was CD45 positive?

There seems to be a significant difference in the number of analyzed cells between murine control and murine tumor tissues in Fig. 4a and 4b. The authors state that "the healthy tissue was always more difficult to mechanically break apart into single cells, and tumor tissue yielded more intact cells". However, from the scatter plots in Fig. 4a and 4b, it appears that many more control cells (= healthy) were analyzed than tumor

cells. Please comment.

In the analysis of murine samples, the authors exclude cells with a size of $60\mu\text{m}^2$ or smaller. However, in the analysis of human samples, all cells are included and analyzed. The authors should also assess the distribution and number of CD45 positive cells in human samples.

The authors state that there are three clusters of cells that correspond to hepatocytes of different sizes, according to the scatter plot in Supplementary Fig. 4b. It is difficult to understand how these clusters were selected. Please indicate the clusters in the figure. Also, explain what the diagonal lines represent and what does the square box indicates?

The authors list papers presenting alternative new and rapid intraoperative methods. There is a paper by Glaser et al. which presents a rapid intraoperative surface microscopy method of fresh breast tissues for guiding surgical oncology. This paper should be cited. *Nat Biomed Eng* (2017) Jul;1(7):0084. doi: 10.1038/s41551-017-0084. PMID: 29750130 entitled "Light-sheet microscopy for slide-free non-destructive pathology of large clinical specimens"

Please use a larger font size in Fig. 4d-g.

Tue 01 Nov 2022

Decision on Article nBME-21-2659A

Dear Prof Guck,

Thank you for your revised manuscript, "Single-cell physical phenotyping of mechanically dissociated tissue biopsies for fast diagnostic assessment". Having consulted with the original reviewers, I am pleased to write that we shall be happy to publish the manuscript in *Nature Biomedical Engineering*.

We will be performing detailed checks on your manuscript, and in due course will send you a checklist detailing our editorial and formatting requirements. You will need to follow these instructions before you upload the final manuscript files.

Best wishes,

Valeria

Dr Valeria Caprettini
Associate Editor, Nature Biomedical Engineering

Reviewer #1:
Report for the authors (Required):
The authors have addressed my concerns. I recommend that the manuscript be published.

Reviewer #2:
Report for the authors (Required):
The authors have significantly improved the manuscript and addressed weaknesses highlighted in the first round of review. In particular they have addressed my first round of comments and now further validate the importance of deformability parameters, by training separate models without these parameters for comparison. The authors also find that size and deformability were independent predictors in the models. Two new supplementary figures are included to present these updated results. In a new supplementary figure 12 the authors also show more data and images of the cryopreserved vs. fresh cells, which indicate that although fewer cells are present, the cells that are present have similar contrast and morphology as fresh cells. This new figure is helpful in supporting the unexpected result that cells from frozen tissues can be analyzed with minimal changes to biophysical properties.

Reviewer #3:
Report for the authors (Required):
The authors have satisfactorily responded to all my comments and revised the manuscript accordingly.

Rebuttal 1

Re: Point-by-point reply to Reviewers for resubmission of manuscript (nBME-21-2659)

“Single-cell physical phenotyping of mechanically dissociated tissue biopsies for fast diagnostic assessment”

Reviewer #1 (Report for the authors (Required)):

The manuscript by Soteriou, et al., provides evidence that solid tissues can be quickly dissociated and analyzed using RT-FDC, and that this approach can be used for diagnostic purposes. Several examples are shown, including inflammatory bowel disease and the ability to distinguish between healthy and cancerous tissue, with the goal of establishing a label-free diagnostic pipeline based single-cell deformability, size and bright-field imaging properties. The RT-FDC achieves very high throughput when compared to typical single-cell biophysical measurements and the integration of deformability with bright-field imaging provides a unique way to assess cell state. A major strength of the manuscript is the inclusion of this approach with a rapid mechanical tissue dissociation to examine clinical specimens.

Although including multiple examples within the manuscript highlights the generality of the measurement platform, it comes at the expense of depth and establishing a compelling use case. Most central to the manuscript is distinguishing between healthy and cancer tissue. However, the scenarios examined involve tissues that are predominately cancer or predominately healthy, which could be identified macroscopically. The real challenge would be to find a small island of tumor (or small fraction of tumor cells) within an otherwise normal specimen or a similar clinical scenario of actual relevance. Thus, the clinical implications based on what is shown in the manuscript are not clear.

We thank the Reviewer for their positive feedback and the constructive criticism. We have on purpose included various tissues to showcase the potential of the method for different applications, even though we have to admit that they are included mostly for illustrative purposes. No single manuscript could be both general and go broadly into depth. We have toned down the claims regarding general performance of physical phenotyping in tissues other than colon in favour of strengthening the specific use case of intraoperative pathological analysis. The central use case presented in this manuscript is indeed the distinction between predominantly cancerous and predominantly healthy tissue. We would like to emphasize, however, that we analyse a very small fraction of the biopsy tissue (50-200 mg) which is selected at random and which is hardly macroscopically identifiable and nearly impossible to be used for conventional histopathological analysis. Importantly, and this is now newly added, the method can also detect small amounts of cancer cells within a large amount of healthy tissue, as suggested by the Reviewer. To support this claim, we have conducted additional experiments and performed an extended analysis of the results.

Specifically, we performed experiments in which we analysed a mixture of tumour and healthy lung tissues (see the new Supplementary Figure 13). Samples containing 50%

tumour and 50% healthy tissue could be identified correctly as tumour. The following text has been added to the manuscript (pages 10-11, lines 306-330, marked in yellow in the manuscript):

We also tested the sensitivity of our approach to the situation where only few cancer cells are present (tumours with low tumour cellularity and extensive desmoplastic tumour stroma content) or remain (tumours following chemo- or radiochemotherapy with nearly complete remission) in the tissue samples available. This aspect is especially important for clinical situations where intraoperative analysis is used to determine whether the operative margin is free of cancer (so called frozen sections), which can be particularly difficult when only very few tumour cells are present. We performed an experiment in which we analyzed a mixture of fresh tumour and healthy lung tissue samples at different ratios (see Supplementary Fig. 13). A sample consisting of 50% healthy and 50% tumour tissue was classified as a tumour sample. Of course, not all the tumour tissue consists of cancer cells, so that the real sensitivity to detect cancer cells is higher than apparent here. In fact, some of the colon tumour samples had relatively high stromal content. In the extreme cases (Supplementary Table 7: frozen colon tissue samples 7 and 11) the stromal content was 98% and 80%, meaning the patients had nearly no residual tumour after neoadjuvant radiochemotherapy. Still, these samples were correctly classified as tumour. This result is remarkable as it points out a possible solution to the sampling problem present in conventional histopathological analysis – especially in frozen section scenarios. The result of the latter very much depends on whether the pathologist inspects and selects the correct tissue location where cancer cells are still present. Due to time constraints and technical limitations of slide preparation in frozen section scenarios only a small fraction of the total resected tissue specimen can be visualized. If the dissociation of the tissue into single cells, and the analysis of a random subset of these, can sensitively detect the presence of as low as 20% – or even 2% – of cancer cells present in a given tissue sample, this would be a clear advance over the state-of-the-art. More specific research is needed to firmly establish this. Finally, our method can also detect low-grade cancer, where any differences in physical parameters are expected to be more difficult to detect than in high-grade cancer. The vast majority of the analysed samples were G2 (moderately differentiated) or G3 (poorly differentiated/undifferentiated; often also referred as “high grade”; Supplementary Table 5-10). In the lung tissue dataset one of the samples was classified as the lowest grade G1 (well differentiated). Therefore, one can appreciate that the method is not limited to high-grade cancer.

Supplementary Fig. 13: Testing of the performance of logistic regression on mixed samples consisting of different ratios of tumour and healthy lung biopsy samples. The percentages indicate the percentage of the volume of the dissociated tumour sample in the total sample volume. The pure tumour sample and the sample with 50% tumour content were classified as tumour, while 25%, 10% and 0% of tumour content resulted as the “healthy” category.

More importantly, the newly added Supplementary Tables 5-10 contain clinical information for the measured human biopsy samples, including stromal content. We note that the measured samples were not purely consisting of cancer cells – some of the samples had relatively high stromal content. In the extreme cases (frozen colon tissue samples 11 and 7) the stromal content was 80% and 98%, since the patients had nearly no residual tumour after neoadjuvant radiochemotherapy. Still, these samples were classified as tumour. This is particularly exciting, as it addresses the sampling problem of conventional histopathological analysis of frozen tissue sections. Since not all areas of all tissue sections can be visually inspected by a pathologist, small amounts of tumour could be missed. This is especially relevant in the intraoperative analysis of tumour margins where only few tumour cells might still be in the sample. Here, the approach of dissociating the tissue, and randomly sampling the individual cells might offer a solution, if it is sensitive enough to detect the presence of as little as 2% tumour cells in the sample. Of course, this result will have to be followed up in clinical trials, but the prospect is exciting and paints a very clear path towards a specific use case of our approach.

In Figure 2, the authors measure physical parameters of cells from murine liver, colon and kidney samples and then examine immunophenotypes for various sections of the scatter plots. In one example they show a cluster of cells with similar physical properties was mainly composed of EpCAM positive cells (Fig. 2a). The authors claim this shows ‘a clean population of epithelial cells can be distinguished in a label-free manner, purely using image-derived physical parameters.’ However examining arbitrary regions of scatter plots of individual samples as is done throughout Figure 2 does not seem useful by itself. It would be more compelling if the authors could define a clear use case and

then show how biophysical parameters leads to superior performance over immunophenotypic markers.

We agree with the Reviewer and are presenting these results in Figure 2 now more as illustrative cases for potential applications, apart from the specific use case of diagnosis of colon inflammation and cancer. We still feel that the inclusion of results from other tissues is of interest to the readership of NBE and might inspire other independent research. We have toned down or eliminated claims of biophysical markers performing better than immunophenotypic markers, since this is unsubstantiated in the present manuscript and distracts from the main point.

Major concerns:

1. The reproducibility of the work remains unclear. In several figures, such as figure 2, the authors display a single example sample, but the reproducibility of the conclusions across independent samples is unclear. Some figures, such as supplemental fig 1, have unacceptably low N numbers. In some figures, such as figure 3, the authors report a higher N number, but whether the N refers to independent experiments or, for example, just technical replicates from the same patient, is not known. Considering the claims that the method should be adapted for routine clinical practice, including more blind-controlled experiments (as was done with a few samples in figure 5a) is essential.

We would like to thank the Reviewer for pointing out concerns regarding the reproducibility of our work. We have performed more measurements of different types of murine tissues to demonstrate the reproducibility of our approach compared to enzymatic protocols. We have updated Supplementary Figure 1 to include the new measurements and updated the caption to include the new 'N' numbers accordingly and updated the Method section and Supplementary Table 1 and 2. We now have a minimum of 4 biological repeats for the tissue grinder and 3 for the enzymatic protocols.

We have now thoroughly inspected the manuscript and updated the N numbers for all figures to correspond to the correct numbers. Throughout the manuscript N number refers to biological repeats.

To strengthen the proposed clinical use case of our platform, we performed 3 additional blind experiments for fresh human colon biopsies and 2 for the fresh lung biopsies, which we have added to Figure 5 as crosses (green for tumour, purple for healthy). We have modified the main text (see page 10, lines 295-303; marked in yellow) accordingly:

Upon logistic regression, only 3 out of 22 samples used for PCA and 1 out of 6 of the blind samples were not correctly classified, which could be attributed to inter-tumour or intra-tumour heterogeneity. Nevertheless, using our approach on blind samples we achieved 100% accuracy in classifying healthy and tumour samples from frozen biopsies, and 83% accuracy for the separation of fresh biopsy samples.

To validate our method on tissue from a different organ, we applied it to freshly excised lung biopsy samples from nine cancer patients. PCA combined with logistic regression readily separated 7 healthy from 7 tumour samples and further 4 blind samples were correctly classified

(Fig. 5c). In the PCA, 46.9% of the variance was explained by PC1 and PC2 (31.2% and 15.7%, respectively).

Fig. 5: Distinction of tumour and healthy tissues in human biopsies using PCA and logistic regression. In the PCA plots on the left, each green point represents a tumour sample from one patient; purple points represent the corresponding healthy surrounding tissue from the same patient. Logistic regression was performed on each of the PCAs with the resulting two categories shown as purple (healthy) and green (tumour) background colours. Crosses represent blind experiments used for the validation of the trained model. The feature importance analysis to the right of the PCA plot shows the colour-coded significance of each feature for determining PC1 and PC2 for that particular tissue; the x axis lists cell size categories; the y axis lists RT-FDC parameters and their statistical features SD (Median) derived across all samples (in brackets). **a**, PCA on RT-FDC parameters of 32 frozen colon samples (16 tumour biopsies and 16 samples of healthy surrounding tissue). **b**, PCA on RT-FDC parameters of 28 fresh colon biopsy samples (14 tumour, 14 healthy). **c**, PCA on RT-FDC parameters of 18 fresh lung biopsy samples (9 tumour, 9 healthy).

2. Given the wide dimensions of disease spectrum, the authors should be more realistic in their discussion of the method potential as probably not all disease will have a strong signal to noise in deformability / imaging space. Without establishing the dynamic range of pathologic deformability vs normal range, it's difficult to assess whether the approach has broad vs. limited use.

We agree with the Reviewer that the presented work does not prove that our method could be applied universally to any kind of cancerous biopsy, the applicability has to be tested on a case-by-case basis. We have edited the Discussion (page 11, lines 366-373; marked in yellow) to give a more realistic view of the method's potential:

PCA of murine colon samples and human colorectal biopsies revealed that cell deformation in standardised channel flow conditions is key for distinguishing between healthy and tumorous tissue in the examined biopsy types. This highlights the uniqueness of the information brought by this method, currently missing from routine diagnostic practices which to date rely mostly on histological assessment. Following this proof of concept study, it will be necessary to investigate whether the method can be adapted to different types of cancer or tissue. We expect that cell deformability changes might manifest more in certain types of cancer than in others. This, there might be certain application areas where the method has a particularly high potential to improve diagnostic practice.

3. The abstract has the strong claim that “[...] distinguish subpopulation of cells..] without prior knowledge or the need for molecular marker”. The authors should elaborate more on the need to eventually or synergistically use molecular markers with the RT-FDC. For example, in Fig.2a it is not clear if the cluster of epithelial (EpCAM positive) cells (having a faintly lighter blue tone and artificially highlighted in green) could be necessarily picked out without relying on the molecular marker.

This is a very good point. Combining RT-FDC parameters with molecular markers will enhance the information obtained for each cell and will allow the user to identify cell populations with greater confidence. This was actually a main point of another one of our recent publications on RT-FDC, where we introduced sorting capabilities (Nawaz et al., Nat. Methods, 2021), which we do not need to elaborate here again. We have amended the discussion (pages 12, lines 398-404, marked yellow) to emphasize the possibility of synergistically using molecular markers and RT-FDC parameters:

Finally, an important aspect of the method is that the physical phenotype of cells can be used to identify cell populations in tissue either in a fully label-free manner or synergistically with molecular markers, enhancing the fluorescence measurements. Furthermore, thanks to the sorting modality recently added to RT-FDC³⁶, a specific population of cells can be isolated according to parameters calculated from images in real-time or using trained neural networks^{64,65}.

Even though not strictly relevant in the present manuscript, we would like to comment on our approach to select clusters in scatter plots of RT-FDC data. The traditional approach for identifying subsets of cells in flow cytometry is to reduce the data set dimensionality using gating strategies based on scattering parameters and molecular marker expression. Here, we adopt a gating strategy based on RT-FDC parameters (such as size, brightness

or deformation) and expression of molecular markers to identify possible subset of cells. In the scatter plots, the marker hue is color-coded according to the kernel density estimate (using the viridis colour map), meaning that the brightest yellow areas are areas with the densest events, while dark purple areas have the lowest density. We apologize for omitting a colour scale, this has now been amended. Currently, a well-separated cluster of events or a cluster with higher density is selected by manual polygon gating, similar to the approach used in conventional flow cytometry. The Reviewer has correctly noticed that when several clusters are rather close to each other, we aim to keep the polygon gates relatively conservative around the densest areas (as seen e.g. in Figure 2a, marked green), aiming for a single clean subpopulation of cells. In the future, we also plan to apply automated clustering algorithms.

4. Related #3, the approach may be stronger when combining fluorescence readout with RT-FDC (Figs. 2b,c + Sup Fig. 3). The authors should consider amending their statement in the abstract by emphasizing enhancing the fluorescence measurement, rather than substituting it for single-cell measurements. Same for conclusion statement (lines 363-365).

In addition to the text changed in response to comment number 3, we have also amended the abstract as follows (marked in yellow):

We show that physical phenotype parameters extracted from brightfield images of single cells can be used to distinguish subpopulations of cells in various tissues, **enhancing or even substituting measurements of molecular markers.**

5. When the authors describe the potential for sorting/studying cell-doublets (line 138-139), the value of a qualitative measure of deformability (larger deviation from aspherical cells discussed in lines 332-334) is not clear. Can orientation be a big confounding factor in such a measurement?

It is important to emphasize that in the case of doublets, a qualitative measure of deformability via RT-FDC is not possible using the current models. In our analytical model and numerical simulations (Mietke et al., Biophys. J., 2015 and Mokbel et al., ACS Biomat. Sci. Eng., 2017), the basic assumption is that the initial shape under normal stress-free conditions is a sphere, which is obviously not the case for cell doublets. Therefore, the measured deformation of a cell doublet cannot be quantified by an apparent elastic modulus. However, our aim here was not to quantify deformability of cell doublets, but rather to use the images to distinguish cell doublets from single cells. For this purpose, the deformation parameter is still useful, as doublets have a higher deviation from circularity in the channel than single cells. However, from our experience, other features derived from images are more useful for the purpose of distinguishing doublets, mainly the aspect ratio and shape features such as Fourier descriptors.

With regards to the orientation, we believe this is rather a benefit of RT-FDC, as the cell doublet long axis aligns with the flow direction when passing through the channel, decreasing the degrees of freedom. Thus, it is in fact easier to distinguish doublets from other cells via feature selection or via machine learning. Furthermore, thanks to this

orientation, it is possible to distinguish fluorescence signals coming from each of the two cells, as they occur sequentially (see Supplementary Figure 3).

6. The “Tissue Grinder – RT-FDC” data on lung tissue (Supp Fig. 4c) are not discussed within the text of paper. Also, it seems that enzymatic dissociation is biased towards lower cell sizes for liver, for lung it is not the case (this is evident in Run 2, while Run 1 has a sudden cut-off). Could the authors comment on why this happens? The current statement in the opening line of the paragraph (line 150) thus seems ambiguous, and the whole paragraph seems not well integrated with the rest of the text and figures.

We agree with the Reviewer that the differences between grinding and enzymatic dissociation were not strongly supported by the presented lung data (Run 1 vs Run 2). We therefore performed three more measurements of enzymatically dissociated murine lungs and compared it to lungs dissociated with a tissue grinder. We observed that larger cells are present in these repeats, therefore the difference between mechanical and enzymatic dissociation is not prominent. We have added this new data to Supplementary Figure 4 and to the result section (page 6, lines 157-159):

In other tissues, such as lung, the differences between mechanical and enzymatic dissociation were not as prominent and neither technique gave a bias towards a specific cell population (Supplementary Fig. 4d).

Supplementary Fig. 4: Physical phenotype characterisation of cells isolated mechanically and enzymatically from murine lung and liver. a, Scatter plots of deformation vs cell size for cells isolated from mouse liver tissue using a tissue grinder (TG) or enzymatic dissociation, showing the enrichment of hepatocytes following mechanical dissociation for 3 independent biological repeats. **b**, Scatter plot of deformation vs cell size showing 3 clusters of cells that correspond to hepatocytes of different sizes; with the corresponding kernel density estimate

(KDE) plot and representative images (r = radius of cells). **c**, Percentage of hepatocytes to the total number of liver cells, as detected by RT-DC for five independent biological repeats. **d**, Scatter plots of deformation vs cell size for cells isolated from mouse lung tissue using a tissue grinder or enzymatic dissociation for 3 independent biological repeats.

We have also modified the text to better integrate the paragraph starting at line 150 (now page 6, lines 143-149 in the revised manuscript) with the other results presented in this section:

An important question to consider when using mechanical dissociation of tissues and label-free analysis by physical phenotype is whether this approach faithfully represents the distribution of cell types present in the tissue. While this is impossible to assess for all tissues and applications in general, it is instructive to have a closer look at liver as a specific tissue (Supplementary Fig. 4a-c). Mechanical dissociation seems less disruptive to sensitive cells such as hepatocytes, which are prone to cell death and often lost during standard isolation procedures³⁸. Upon dissociation of murine liver tissue, cells above $150 \mu\text{m}^2$ in cross-sectional cell area (ca. $7 \mu\text{m}$ radius) were determined as hepatocytes according to their morphology and size³⁹.

7. In figure 2, the authors present a range of phenotypes that they can detect. The authors should address if these phenotypes reflect diverse cellular responses to the TG or RT-FDC, rather than phenotypes originally present in vivo. Furthermore, on line 157, the authors say that “using TG, the proportion of hepatocytes to total cells was between 40-80%”. The authors should justify this result and provide evidence for whether this is a purely a technical artifact or a biologically meaningful result.

The Reviewer has a good point, cellular responses due to sample processing may indeed enhance the heterogeneity of the detected physical phenotypes. Although this knowledge is not necessary for the diagnostic purposes that are central to the manuscript, is certainly an interesting and important topic to address in our future experiments. We have amended the relevant paragraph in the Results section, as shown below.

Regarding the Reviewer’s point about hepatocytes, we know that the spread in hepatocyte percentages obtained via tissue grinder is not an exact representation of the percentage of cells present in the tissue, but rather influenced by technical aspects, as the percentage of hepatocytes in the liver is reported to be 60-70%. We performed five biological repeats and included the hepatocyte percentages for each of the repeats in the new Supplementary Figure 4 (see above), as well as the average and standard deviation. The actual percentage obtained with the tissue grinder on average is about 50%, which is much closer to the real value than enzymatic dissociation (less than 10%) in our hands. We have accordingly adjusted the main text (page 6, lines 149-157):

As the major parenchymal cell type of the liver, hepatocytes account for 70% of the liver cell population and take up nearly 80% of liver volume⁴⁰. In the cell suspension obtained using TG, the proportion of hepatocytes to total cells was on average 52.5%, much closer to the real representation in tissue compared to the 7.7% for enzymatic digestion. Moreover, distinct subpopulations of hepatocytes could be identified according to cell size. We hypothesize that these populations correspond to hepatocytes of differing ploidy, as DNA content is strongly correlated with cell volume⁴¹. If confirmed, for example by correlation with a quantitative fluorescence analysis of DNA amount in each cell, our method could serve as a tool for the

label-free monitoring of aging and pathophysiological processes in the liver, which are linked with the proportion of polyploid hepatocytes⁴².

Minor comments:

1. What is the high-viscosity medium in line 109? Are there special requirements that need attention? Please specify.

Indeed, the buffer is composed of methyl cellulose in PBS with carefully adjusted viscosity and osmolality. The composition of the high viscosity medium is detailed in the Methods in the “Tissue dissociation and single cell preparation” section, as follows (page 14, lines 453-457):

The cell pellet was resuspended in a high viscosity measurement buffer prepared using 0.6% (w/w) methyl cellulose (4,000 cPs; Alfa Aesar) diluted in phosphate buffer solution (PBS) without calcium and magnesium, adjusted to an osmolality of 270-290 mOsm/kg and pH 7.4. The viscosity of the buffer was adjusted to (25±0.5) mPa·s at 24 °C using a falling sphere viscometer (HAAKE Falling Ball Viscometer Type C, Thermo Fisher Scientific).

For clarity we have also specified the buffer in the main text (page 4, line:103)

In an RT-FDC measurement, hundreds of cells per second, suspended in a high-viscosity methyl cellulose buffer,

2. Line 128: In text it is stated that there are 6 clusters of cells in Fig. 2b while in the figure there are 7 clusters displayed.

Thank you for pointing out the mistake, it has been amended.

3. It would help to remove various faint horizontal and vertical lines in Figure 3 and make sure that plots are consistently presented as closed or open boxes.

We believe this was an artefact of the PDF, as the thin lines are not there in our vector graphics files. We have now inserted them into the PDF as a different format and will make sure to check this artefact when uploading the individual vector graphics files for final submission.

We have removed the closed box format of Figure 3e to make the plots more consistent.

4. Figs. 3c,4c: include legend or text in caption to explain how the margins are drawn.

We apologize for having omitted a proper explanation of Fig 3c, which shows kernel density estimate plots. The caption now includes the following text:

Kernel density estimate plots of samples shown in A and B, with contours marking the 0.5 (light shade, outer contour) and 0.95 (dark shade, inner contour) levels.

5. Generally Supp Fig. 4, if included, should have clear markings of the cell images as to where they belong in the RT-FDC graphs as it is done for example in Fig. 2b,c.

That is a good suggestion; we have edited Supp Fig. 4 accordingly, as shown in our reply above.

6. For consistency, in line 184, change “number” to “percentage” to be consistent with Fig. label on horizontal axes of Fig. 3d,e.

Thank you for the suggestion, we have changed this in the text.

Reviewer #2 (Report for the authors (Required)):

The manuscript describes the application of high-throughput microfluidic single-cell physical phenotyping to solid tissue biopsies towards applications in diagnosis of diseased states in tissues. This is an important topic since physical phenotyping approaches can enable rapid and unbiased testing that allows, for example, measurement of malignancy during surgical procedures, rather than after histological sectioning, staining and analysis. Previous work in the field has mainly addressed cells naturally in suspension like in blood or other body fluids. The key breakthrough of this work is showing the ability to use a mechanical dissociation approach to obtain single cells compatible with the high-throughput microfluidic measurements. It appears that the approach to dissociation is also robust and repeatable enough to identify disease-specific signatures of cells across many samples. Several disease models are studied, including controlled animal models of colitis and malignancy, as well as human tissues (including previously frozen tissues). Overall, the work is well conducted and reported on an important and impactful topic. Some of the claims should be better substantiated with data analysis approaches. There should also be more clarity on the classification of the clinical disease states and what samples were used to train vs. validate the accuracy of the test.

We thank the Reviewer for the very favourable opinion about our work, its importance, relevance and timeliness, and for their valid recommendation to further substantiate some of our claims with further analysis. We have extended the data analysis according to the specific comments of the Reviewer, e.g. by adding validation data for human biopsy classification and by training a model after removing key deformation parameters. We hope the Reviewer considers these additions to have improved the manuscript.

Specific Comments:

Major:

Many claims throughout the manuscript point to the added value of parameters derived from cell deformation on disease diagnostic accuracy, however these parameters were not directly evaluated as useful compared to related parameters. For example, in the transfer colitis model, can the authors determine if the deformation parameters provided additional information beyond the amount or fraction of infiltrating immune cells which could be obtained using other processes (e.g. hematology analyzer following mechanical tissue dissociation). In the mouse tumor model, was there any correlation between deformation and size (and the statistics of these metrics) or were these parameters independent predictors of being a tumor sample? Understanding the correlation between the different parameters would be helpful to better elucidate what leads to the differential signal. Another way to test the importance of the deformation parameters is to train a model that does not use them and see how well that model performs. If you remove deformation parameters, can you still perform PCA to separate

the samples for the human biopsy studies? How well does this perform?

We thank the Reviewer for these questions and suggestions. We believe the added value of the cell deformation parameters was already evident from the distinction between healthy and tumour tissue samples, both for murine and human tissues. This can be observed in Supplementary Figure 7 and in Figure 5, where the colour coded feature importance diagram demonstrates that the deformation parameters have high scores compared to the other features used.

However, we considered it a great idea to train a model leaving out deformation parameters to see how their absence would affect the principal component analysis and classification. We have done so for the human colon frozen biopsies, where the performance of the algorithm with deformation parameters included was excellent. We left out three deformation parameters which had the strongest contribution to the PCA, according to the parameter importance scoring: deformation mean, median and SD of cells of size 100 – 400 μm^2 . PCA without these parameters resulted in worse separation between healthy and tumorous tissue (see new Supplementary Figure 10b below) than the original (Suppl. Fig. 9a below); 6 out of 32 samples ended up misclassified (18.75%). We have now included this result as a new Supplementary figure 10 and added a sentence to the Results section (page 9, lines 264-268; marked in yellow):

The PCA showed that tumour and healthy samples segregated well along PC2 mainly by the deformation and standard deviation of brightness of cells larger than 100 μm^2 (Fig. 5a). Cell size parameters of cells below 100 μm^2 also contributed to the separation of the samples. Excluding the most important parameter (deformation of cells larger than 100 μm^2) resulted in worse separation between healthy and tumour samples (Supplementary Fig. 10).

Supplementary Fig. 10: Testing the performance of healthy vs tumour classification of frozen human colon samples. **a**, PCA and logistic regression including the full set of 45 parameters as input for PCA; **b**, PCA and logistic regression after excluding 3 parameters (deformation mean, median and SD for cells larger than 100 μm^2).

We also considered the Reviewer's question about the mouse tumour model and potential correlations between deformation and size and the statistics of these metrics. The Pearson's correlation coefficients for deformation and area for all events are shown in the

newly added Supplementary Fig.8 below. The correlation coefficients for the statistical means and medians of the samples (divided into the 3 cell size categories used for PCA) are shown in the table below. The values were either not correlated, or the correlation was weak. Therefore, we conclude that deformation and cell size were independent predictors in the tumour vs healthy classification of murine samples.

The following text was added to the manuscript (page 8-9, lines 250-256; marked in yellow):

Logistic regression performed on the PCA (shown by the linear divide in Fig. 4h) demonstrated that the condensed physical phenotype information represented by the principal components suffices to distinguish between healthy and tumour tissue; 29 out of 32 samples lay in the correct region. Finally, we analysed the correlation between deformation and cell size and found it to be weak or non-existent (Supplementary Fig. 8). This led us to conclude that deformation and cell size were independent predictors of tumours in murine colon samples, further demonstrating the added value of deformation measured via RT-FDC as a diagnostic marker.

Supplementary Fig. 8: Correlation of deformation and cell size in murine healthy and tumour samples. a, Pearson's correlation coefficients for deformation and area; each point corresponds to one murine sample and b, table showing Pearson's correlation coefficients for the statistical means and medians of area and deformation, divided into the three cell size categories used for the principal component analysis.

For the fresh tumor biopsies it was unclear what the training / validation sets were for these experiments when reading the main text. If there were not separate validation data sets, this should be clearly indicated.

We have now included validation data (blind experiments) also for the fresh colon and fresh lung experiments shown in Figure 5 (6 samples and 4 samples, respectively). These data are shown as crosses in Fig.5 b, c. This is also clarified in the caption (marked in yellow):

Fig. 5: Distinction of tumour and healthy tissues in human biopsies using PCA and logistic regression. In the PCA plots on the left, each green point represents a tumour sample from one patient; purple points represent the corresponding healthy surrounding tissue from the same patient. Logistic regression was performed on each of the PCAs with the resulting two categories shown as purple (healthy) and green (tumour) background colours. Crosses represent blind experiments used for the validation of the trained model. The feature importance analysis to the right of the PCA plot shows the colour-coded significance of each feature for determining PC1

and PC2 for that particular tissue; the x axis lists cell size categories; the y axis lists RT-FDC parameters and their statistical features derived across all samples (in brackets). **a**, PCA on RT-FDC parameters of 32 frozen colon samples (16 tumour biopsies and 16 samples of healthy surrounding tissue). **b**, PCA on RT-FDC parameters of 28 fresh colon biopsy samples (14 tumour, 14 healthy). **c**, PCA on RT-FDC parameters of 18 fresh lung biopsy samples (9 tumour, 9 healthy).

This Reviewer was surprised that intact cells were observable from cryopreserved tissue sections, since there would be cell death and cell membranes compromised in this process. How does this affect the images/contrast? One would expect contrast would be decreased substantially given the ability of the suspending solution to transfer into the cell if the membrane is disrupted. Can the authors show images of cells observed from these various tissue sources to understand how the differences in cell images, initial circularity, and contrast, may affect the performance of the technique.

Like the Reviewer, we were positively surprised with the performance of the method for frozen colon tissues. Supplementary Figure 12 compares the same biopsy sample analysed by RT-FDC in its fresh state and following rapid freezing in liquid nitrogen. We have updated the figure to include representative images of cells from each sample. The quality of images of cells detected as such in the frozen samples appears to be comparable to those from fresh samples. However, dead cells or cells with compromised membrane integrity, which will definitely be present in the sample after freezing, may not be detected as events due to their low contrast. This is likely also the reason for the lower quantity and heterogeneity of cells upon freezing, which is evident in the scatter plots in Supplementary Figure 12. However, what is important is that there seem to be sufficient events with good quality in order to achieve excellent classification for frozen tissue samples (Figure 4).

Supplementary Fig. 12: Comparison of physical phenotype parameters of cells from frozen and fresh human biopsy samples. Cell size vs deformation scatter plots of single cells extracted from either fresh (purple) or frozen (green) colon biopsy samples; **a**, healthy sample; **b**, tumour sample; including 3 sample cell images for each plot with a scale bar = 10 μm^2 . The kernel density estimate (KDE) plots on the right correspond to the scatter plots on the left; the histograms show the distributions of cell size and deformation. **c**, Medians and standard deviations of cell size,

deformation, area ratio and aspect ratio of fresh ($N = 6$) and corresponding frozen ($N = 6$) samples. Boxes extend from the 25th to the 75th percentile with a line at the median; whiskers span 1.5x the interquartile range. Statistical comparisons were performed using Wilcoxon signed rank test, cell size SD ($*p = 0.0277$, $r = 0.64$), median area ratio ($*p = 0.0277$, $r = 0.64$) and area ratio SD ($*p = 0.0277$, $r = 0.64$); r : effect size; ns: non-significant.

The authors state “The speed of the extraction presumably helps to preserve biochemical and biophysical phenotypes in conditions close to those in situ.” It is unclear what evidence supports this statement. A main concern would be how mechanical shearing affects the physical properties of cells and whether some populations are damaged more or are more adhesive to other cells and stay in aggregates. Then these populations would be underrepresented in the final analysis process. Perhaps the authors could evaluate the cell type distributions in the tissue using staining approaches and compare to the cell types that make is through to single cells for analysis to determine if there is any bias / enrichment effect. I understand this information does not necessarily impact the classification results reported as it may be that differences in enrichment/depletion is part of the signal observed that leads to the differences between diseased and normal tissue.

The Reviewer has a valid point. Mechanical shearing may affect the physical properties of cells, which will likely be different to the properties in situ. One cannot simply dissociate cells from tissues into a single cell suspension and claim that the measured mechanical properties and cell representation are the same as in its original location (which holds true for any type of dissociation, not only mechanical). It is also possible that certain populations of cells are more sensitive than others to damage done by the dissociation protocol; this is in fact already known to be the case for certain enzymatic protocols (Waise et al., Sci. Rep., 2019). The investigation of effects of mechanical dissociation via tissue grinder on physical phenotypes and cell population enrichment is certainly a topic for a large separate study. Such effects will be very specific for each tissue type and for each subpopulation of cells, we are therefore facing a very large and demanding experimental project. We think that the best approach will be to do single cell sequencing and physical phenotyping in parallel, which is technically a challenge. Like the Reviewer, we also consider it an intriguing and very important question and are currently working on setting up such a system.

However, the main focus of this current manuscript is the distinction of a disease vs. healthy states, rather than a detailed characterisation of the changes upon mechanical dissociation, as the Reviewer acknowledges. For this reason and the extensiveness of the experimental work that would be required, we think it is outside of the scope of this manuscript to investigate to which degree mechanical dissociation affects the physical properties of cells and the enrichment of different populations in different tissues. Even though we present no evidence, we still think that it is conceivable and likely (“...presumably...”) that the speed of the dissociation helps to preserve biophysical and biochemical properties, as we also have argued in the Discussion (page 12; lines 385-389):

Fast dissociation also has the potential to preserve biochemical and biophysical properties of cells in a state near to *in situ*; properties which likely deteriorate with longer processing times in other approaches. Due to the speed of the mechanical dissociation, cells might undergo less proteomic or transcriptional changes, which are known to happen during enzymatic processing^{34,57-60}. Further comparative and molecular studies are necessary to assess these assumptions.

Minor:

In Supplementary Fig. 1 – blue bars are labeled as enzymatic dissociation or mechanical dissociation. The caption indicated it should be enzymatic dissociation. The discrepancy should be corrected.

This is actually not an error; the mesenteric lymph nodes, spleen and thymus were dissociated by mashing the tissue between the frosted ends of two microscope slides, as detailed in Supplementary Table 1 (a mechanical method standardly used for these tissues). This method was used for comparison with tissue grinding. The other organs were dissociated enzymatically, as detailed in Supplementary Table 1. We thank the Reviewer for pointing this out and to avoid confusion to future readers we have updated Supplementary Figure 1 that now reads ‘Standard Dissociation’.

Reviewer #3 (Report for the authors (Required)):

This article by Despina Soteriou et al. entitled “Single-cell physical phenotyping of mechanically dissociated tissue biopsies for fast diagnostic assessment” presents a diagnostic pipeline for the rapid, label-free analysis of biopsy samples by sequentially assessing the physical phenotype of singularized, suspended cells in high-throughput. The authors demonstrate the potential of their method for inflammatory bowel disease diagnostics. Their results suggest that mechanically dissociating tissue biopsies can be used to accurately distinguish between healthy and tumor tissue in mouse and human biopsy samples. The authors claim that their method is quick and delivers results within 30 minutes, laying the groundwork for a fast and marker-free diagnostic pipeline to detect pathological changes in solid biopsies. The manuscript is well-written, and the figures clearly show workflows and results. However, I have several specific questions that I want the authors to address.

We are glad that the Reviewer appreciates our work and we thank them for the specific suggestions, which allowed us to improve the manuscript.

Specific comments

My main concern with this paper is the mechanical grinder method because it is harsh to the cells and should cause significant cell death. Cells are sticky and attach quite firmly to each other and therefore need a relatively large force to separate. Thus, enzymatic treatments are used so frequently. The authors must provide more robust evidence that the mechanical method is similar to or superior to the enzymatic methods. Without further evidence, this method lacks generality and can only be applied to a limited number of tissues.

We agree with the Reviewer’s concerns that excessive mechanical force can indeed lead to cell death. However, the tissue grinder is not as harsh as the Reviewer suggests. Evidence is given in Scheuermann, et al., Curr. Dir. Biomed. Eng. (2019) and Scheuermann et al., bioRxiv (2021) where they tested the effect of the tissue grinder on cell viability of resected tumour tissues and assessed expression of apoptotic markers following cell dissociation. When comparing the tissue grinder to explant procedures they observed that the tissue grinder was more efficient in isolating cells from tumour biopsies. The authors concluded that processing with the tissue grinder resulted in higher viability, cell yield and heterogeneity of the cells compared to the commonly used explant method.

In addition, our data presented here also supports the notion that the mechanical grinder method is not harsher to cells and does not cause higher cell death than conventional protocols. In our experiments, the cell yield and viability observed for tissues processed using a tissue grinder were comparable to the conventional dissociation methods (Supplementary Figure 1). In addition, the tissue grinder method led to decent recovery of hepatocytes, which are extremely sensitive cells that are difficult to isolate. In fact, we were able to recover about 50% of hepatocytes from liver tissue, as opposed to less than 10% with enzymatic dissociation, while literature values report 60 – 70% hepatocytes

(see Supplementary Fig. 4 and the discussion of a similar point by Reviewer #1 above). This leads us to believe that the tissue grinder is actually relatively gentle to cells.

Overall, we believe that our data and the conclusions of Scheuermann et al. are sufficient to state that the mechanical dissociation method is at least comparable to enzymatic methods in terms of the harshness but has an unbeaten advantage due to the simple and quick preparation. In the future, we plan to investigate the proteomic and transcriptional changes caused by tissue grinder and compare them to enzymatic processing, where they are known to happen.

We would also like to argue against the statement that the method can only be applied to a limited number of tissues. In our manuscript alone we present a large number of examined samples where the tissue grinder method proved to be fully functional, ranging from various murine organs to inflamed colon samples, as well as frozen and fresh human biopsies from different organs. So far, we have successfully processed tissues from spleen, thymus, lymph nodes, liver, kidney, small intestine, colon, lung, stomach, and pancreas.

And finally, the main use case of the combination of tissue grinder and RT-FDC presented in this manuscript is the diagnosis of tissues. Here, the individual performance of the tissue grinder does not even matter, as long as the cells it produces can be analyzed, and the results are robust and diagnostic. This could even include a differential mechanical sensitivity of diseased-state vs healthy cells in the tissues processed using the tissue grinder. This has now been discussed more extensively in the revised manuscript (see page 6, lines 162-190).

In Supplementary Fig. 1, the authors compare the cell viability and cell yield between mechanical dissociation using a tissue grinder and enzymatic dissociation using conventional enzymes. The cell viability is assessed by propidium iodide and the cell yield is evaluated by RT-FDC. However, RT-FDC is not the standard method for determining cell viability and yield. The authors must perform similar experiments using FACS.

As in conventional flow cytometry, RT-FDC measures fluorescence signals from each cell passing through the microfluidic channel, in addition to the physical characteristics of cells (Rosendahl et al., Nat Methods, 2018). As such, RT-FDC is also a fluorescence-based flow cytometer with similar performance as standard FACS. The main differences between conventional flow cytometry and RT-FDC are the lower cell throughput and fewer (3) fluorescence channels of the latter. Neither of these differences affect the cell viability measurement using propidium iodide. However, to ensure reproducibility of our data we have repeated these experiments and performed Trypan Blue exclusion assay together with the propidium iodide. We have updated Supplementary Figure 1 accordingly (see below).

The cell yield per mg of tissue upon grinding was in fact not analysed via RT-FDC, but using a conventional cell counter, which is a standard method. This has now been specified in the methods (page 13, lines 422-424).

A. Cell viability assessed with propidium iodide

B. Cell viability assessed with trypan blue

C. Total number of cells per mg of tissue

Supplementary Fig. 1: Comparison of cell viability and cell yield of mechanical vs standard dissociation of different murine tissues. a, Percentage of viable cells for different organs dissociated using a tissue grinder (TG; marked in red) or standard dissociation (marked in blue).

Cell viability was assessed using propidium iodide and RT-FDC. **b**, Number of cells (obtained using a cell counter device) per mg of tissue processed. Lung, liver, kidney, pancreas and stomach processed with enzymatic dissociation were not weighted prior to the experiments. The line represents the mean and the box extends from minimum to maximum values. The number of biological repeats for each tissue is specified in Supplementary Table 1 (standard dissociation) and Supplementary Table 2 (tissue grinder dissociation).

The pathological data of human samples used in this study is not presented. The authors need to show that their method works for both high-grade and low-grade cancers. In high-grade cancer, it is easier to detect differences in physical parameters, such as cell size; however, the differences in physical parameters are much more difficult to detect in low-grade cancer.

This is an important comment and a grave omission of the original submission. Like the Reviewer, we would expect low-grade cancer to be more difficult to detect and more prone to misclassification. In the new Supplementary Tables 5-10 we present the pathological data of the human samples presented in this study. The table also shows the cancer grade. The vast majority of the analysed samples were G2 (moderately differentiated) or G3 (poorly differentiated); no high-grade G4 (undifferentiated) samples were analysed. In the lung tissue dataset, one of the samples was classified as the lowest grade G1 (well differentiated). Therefore, one can appreciate that even low-grade cancer samples can be detected using our method. The following text was added to the Results section (page 10-11; lines 325-330):

Finally, our method can also detect low-grade cancer, where any differences in physical parameters are expected to be more difficult to detect than in high-grade cancer. The vast majority of the analysed samples were G2 (moderately differentiated) or G3 (poorly differentiated/undifferentiated; often also referred as “high grade”; Supplementary Table 5-10). In the lung tissue dataset one of the samples was classified as the lowest grade G1 (well differentiated). Therefore, one can appreciate that the method is not limited to high-grade cancer.

Furthermore, we would like to refer to reply to Reviewer 1 (on page 1-3) where we explained how we tested the sensitivity of the method for detecting small fractions of cancer cells in a sample, and the new Supplementary Figure 13. Even tumour samples with a high stromal content (as high as 98% and 80%) could be reliably classified as tumour. This clearly demonstrates sufficient sensitivity for the intended use case of intraoperative pathological assessment, specifically addressing the sampling problem of conventional histopathological inspection of frozen tissue sections.

In Fig. 5, human cancer and control tissues are used. Were matched pairs used? The numbers suggest that control tissues were taken from the cancer patients, e.g., 13 frozen colon tumors vs. 13 frozen colon control, 11 fresh colon tumors vs. 11 fresh colon control, and 7 fresh lung tumors vs. 7 fresh lung control. The authors must show

how well their method can detect differences in cancer and control tissue from the same patient. This is essential since this is how the method is suggested to be used.

We agree with the Reviewer that it is crucial to show that our method can distinguish differences between healthy and tumour tissue from the same patient. Indeed, matched pairs were used in our experiments. For each patient in Fig.5, a control was taken from the healthy tissue surrounding the tumour. This has now been clarified in the caption of Fig. 5:

In the PCA plots on the left, each green point represents a tumour sample from one patient; purple points represent the healthy surrounding tissue from the same patients.

We have also clarified the Methods section (page 13, lines 436-438):

Matched pairs of samples were analysed, with two samples derived from each patient: a tumour sample and a control sample originating from healthy tissue surrounding the tumour.

In Fig. 2a (liver), the authors show that a subpopulation of cells forms a cluster at a specific cell size and brightness and that these cells are uniform in the enrichment of CD31, CD45, and EpCAM. Why was this cluster chosen? How about other subpopulations from the same scatter plot? Do they have different enrichments of CD31, CD45, and EpCAM?

We are aware that the potential use of our approach, to identify and analyze subpopulations of cells in tissues by their physical phenotypes presents an intriguing option. In fact, we are pursuing this aspect in detail in several projects. However, as prompted by the editor and other Reviewers, for this manuscript, we instead decided to focus on a specific clinical use case – intraoperative diagnostic assessment of colon tissue biopsies – at the expense of providing depth to other uses. As such, we show the results of Figure 2 as illustrative examples of other tissue types and only hint at possible uses (sorting of such subpopulations for further identification and OMICS analysis, etc.). To specifically answer the questions of the Reviewer: the particular subpopulation was chosen rather randomly. The other subpopulations have not systematically been analyzed for their enrichment of these fluorescence markers. However, from our experience with these data, and from other experiments with heterogeneous populations (see for example, Nawaz et al., Nat. Methods, 2021), we can say that subpopulations identified by biophysical phenotyping often also represent unique populations when queried biochemically. One way to show this is to sort out these subpopulations one-by-one and then performing subsequent RNA sequencing, for example. This is outside the scope of the present manuscript. Even though this aspect of the manuscript has not been elaborated in any depth, we still feel that the inclusion of this figure is of benefit to the readers of NBE. However, we are open to removing this Figure entirely should this Reviewer and the editor think it would be appropriate.

The manuscript shows that RT-FDC can detect cell doublets from a cell suspension from murine thymus and spleen. There are many lymphocytes in the thymus and

spleen, and they are relatively uniform in their shape, so cell doublets can easily be detected using RT-FDC. The authors should also show that cell doublets can be detected in other solid organs, such as the colon and kidney.

Thank you for the comment. It is possible to detect doublets also in other organs. We show an example of a doublet in Fig. 2b in murine colon tissue. In addition, we have added an example from the kidney to Supplementary Figure 3. We initially chose the spleen and thymus because the doublet populations were very strong in these organs, which we believe is linked with their immune functions. As can be seen in Supplementary Figure 3e, the doublets are scarce in the kidney.

We note that the distinction of doublets based on the cell size and aspect ratio is only a basic way to do so which should be improved, e.g. by calculating “shape features” (such as Fourier descriptors) or by training a deep learning algorithm to distinguish doublets from single cells. This is subject of currently ongoing activities.

Supplementary Fig. 3: Detection of cell doublets using RT-FDC. Representative scatter plots of aspect ratio vs cell size of cells isolated from murine **a**, thymus, **c**, spleen and **e**, kidney showing the gating strategy for identifying cell doublets. Cell doublets identified in **b**, thymus and **d**, spleen and **f**, kidney with corresponding fluorescence traces, showing a leukocyte (CD45) attached to an endothelial cell (CD31), or the interaction of two leukocytes.

In Supplementary Fig. 6, the authors show that a high percentage of CD45 positive cells are smaller in size than $60 \mu\text{m}^2$. These smaller cells are excluded from the analysis as these cells are mainly “immune cells and small debris”. Please back up this exclusion with experimental data or at least a reference. Of all excluded cells ($<60\mu\text{m}^2$), what percentage was CD45 positive?

We have added percentage calculations to back up our statement that the applied size gate removed mostly debris and leukocytes. Averaging over all analysed samples, debris accounted for nearly 90% of events $< 60 \mu\text{m}^2$ and the percentage of the remaining 10%

of cells that were positive for CD45 was about 57%. In the murine control vs tumour samples, small cells $< 60 \mu\text{m}^2$ were identified by additionally gating for area ratio 1 – 1.05 and aspect ratio 1 – 2. Any events outside of these gates and all events $< 25 \mu\text{m}^2$ were considered as debris. CD45+ cells were identified by fluorescence. Based on this, we added the following percentages to Supplementary Figure 6:

	Mean	STD
debris / all events $< 60 \mu\text{m}^2$	89.6%	10.5%
CD45+ cells / total cells $< 60 \mu\text{m}^2$	57.2%	19.0%

There seems to be a significant difference in the number of analyzed cells between murine control and murine tumor tissues in Fig. 4a and 4b. The authors state that “the healthy tissue was always more difficult to mechanically break apart into single cells, and tumor tissue yielded more intact cells”. However, from the scatter plots in Fig. 4a and 4b, it appears that many more control cells (= healthy) were analyzed than tumor cells. Please comment.

We would like to thank the Reviewer for pointing this out. This sentence is in fact wrongly placed and it was intended for the transfer colitis model as well as the human biopsy samples. In the transfer colitis model, the healthy tissue is more difficult to dissociate than the inflamed tissue. We also observed that the healthy biopsy sample was also more difficult to process than the tumour sample. We suspect that the altered tissue architecture of the tumour might not be as stable as the complex structure of the healthy tissue organs.

We have moved the sentence to page 6; lines 188-190 and modified it accordingly to clarify that this was an experimental observation:

A noteworthy observation was that the healthy tissue was more difficult to mechanically break apart into single cells than the diseased tissue which yielded more cells/events for analysis.

Indeed, in Fig 4a and 5b there are more control cells in the healthy sample than in the tumour sample. The reason behind this observation is that the tumours used for these experiments were very small and hence did not yield a great number of cells/events for analysis.

In the analysis of murine samples, the authors exclude cells with a size of $60\mu\text{m}^2$ or smaller. However, in the analysis of human samples, all cells are included and analyzed. The authors should also assess the distribution and number of CD45 positive cells in human samples.

In the human samples, CD45 positive signal was also common in larger sized events, in contrast to the murine tissue where it was more localised to small cells. We therefore decided to try PCA without excluding any of the cells. The performance turned out to be very good and we kept it that way. We have not added the analysis of CD45 positive cells

to the manuscript, as it does not seem relevant to the results, which were obtained by analysing all cells. We would like to show the Reviewer the typical distribution of CD45+ events on two examples of fresh human tissues, a healthy sample and a tumour sample (left are all cells, right filtered for CD45+ signal). However, we would like to state that some of these CD45+ events might be due to unspecific binding of the antibody.

Ex.1 - healthy

Ex. 2 - tumour

All cells

CD45+cells

The authors state that there are three clusters of cells that correspond to hepatocytes of different sizes, according to the scatter plot in Supplementary Fig. 4b. It is difficult to understand how these clusters were selected. Please indicate the clusters in the figure. Also, explain what the diagonal lines represent and what does the square box indicates?

The Reviewer is right that the identification of hepatocyte clusters according to size should have been made clearer. We have amended Supplementary Figure 4 to better show the distribution and density of hepatocytes. We have also added a kernel density estimate (KDE) plot for this sample (Supplementary Fig. 4c). The clusters should be more evident

now in the scatter plot, where density is indicated by the colour scale, as well as in the contour (KDE) plot.

As per suggestion of Reviewer #2, we have also removed the confusing boxes and replaced them with a line connecting an example each of a hepatocyte from the three densest regions and its coordinates in the plot.

Part of Supplementary Figure 4: b, Scatter plot of deformation vs cell size showing 3 clusters of cells that correspond to hepatocytes of different sizes; with the corresponding kernel density estimate (KDE) plot and representative images (r = radius of cells).

The authors list papers presenting alternative new and rapid intraoperative methods. There is a paper by Glaser et al. which presents a rapid intraoperative surface microscopy method of fresh breast tissues for guiding surgical oncology. This paper should be cited. Nat Biomed Eng (2017) Jul;1(7):0084. doi: 10.1038/s41551-017-0084. PMID: 29750130 entitled “Light-sheet microscopy for slide-free non-destructive pathology of large clinical specimens”

Thanks for alerting us to this interesting and relevant publication. We have added this reference in the Introduction (page 2, lines 61-63):

Moreover, sample preparation is time-, resource- and labour-intensive. Alternative workflows have been proposed²⁸, including stimulated Raman spectroscopy^{29,30}, optical coherence tomography³¹ and **fluorescence microscopy^{32,33}**, but have not yet been implemented.

Please use a larger font size in Fig. 4d-g.

We have increased the font size.